# Adaptive and Robust Watermark for Generative Tabular Data

## Abstract

Recent development in generative models has demonstrated its ability to create high-quality synthetic data. However, the pervasiveness of synthetic content online also brings forth growing concerns that it can be used for malicious purpose. To ensure the authenticity of the data, *watermarking* techniques have recently emerged as a promising solution due to their strong statistical guarantees. In this paper, we propose a flexible and robust watermarking mechanism for generative tabular data. Specifically, a data provider with knowledge of the downstream tasks can partition the feature space into pairs of $(key, value)$ columns. Within each pair, the data provider first uses elements in the $key$ column to generate a randomized set of "green" intervals, then encourages elements of the $value$ column to be in one of these "green" intervals. We show theoretically and empirically that the watermarked datasets (i) have negligible impact on the data quality and downstream utility, (ii) can be efficiently detected, (iii) are robust against multiple attacks commonly observed in data science, and (iv) maintain strong security against adversary attempting to learn the underlying watermark scheme.

## 1 Introduction

With the recent development in generative models, synthetic data (Jordon et al., 2022) has become more ubiquitous with applications ranging from health care (Gonzales et al., 2023) to finance (Assefa et al., 2020; Potluru et al., 2023). Among these applications, synthetic data can be an alternative option to human-generated data due to its high quality and relatively low cost. However, there is also a growing concern that carelessly adopting synthetic data with the same frequency as human-generated data may lead to misinformation and privacy breaches. Furthermore, as modern large-scale language models requires a vast corpus of data for training, there is an urgent need from both researchers and practitioners to prevent "model collapse", where models repeatedly trained on low-quality AI-generated data exhibit gradual degradation in performance (Shumailov et al., 2024). Thus, synthetic data needs to be detectable by any upstream data-owner.

Watermarking has recently emerged as a promising solution to synthetic data detection with applications in generative text (Kirchenbauer et al., 2023; Kuditipudi et al., 2024), images (An et al., 2024), and relational data (He et al., 2024; Zheng et al., 2024). A watermark is a hidden pattern embedded in the data that may be indiscernible to an oblivious human decision-maker, yet can be algorithmically detected through an efficient procedure. The watermark carries several desirable properties, notably: (i) **fidelity** - it should not degrade the quality and usability of the original dataset; (ii) **detectability** - it must be reliably identified through a specific detection process; (iii) **robustness** - it should withstand manipulations from an adversary (Katzenbeisser & Petitcolas, 2000; Atallah et al., 2001); and (iv) **security** - it should be hard for an adversary to learn the exact parameters of the underlying watermarking scheme.

Applying a watermark to the synthetic tabular dataset is particularly challenging due to its structure. A tabular dataset must follow a specific format where each row contains a fixed number of features, i.e., precise information about an individual. Hence, even perturbation of a subset of features in the data can substantially affect the performance of downstream tasks. Furthermore, tabular data is commonly subjected to various methods of data manipulation by the downstream data scientist, e.g., feature selection and data alteration, to improve data quality and enable efficient learning. While prior works (He et al., 2024; Zheng et al., 2024)

| Key A | Value B | Value A | Key B |
|-------|---------|---------|-------|
| $K_1$ | . . . | $V_1$ | . . . |
| $K_2$ | . . . | $V_2$ | . . . |
| $K_3$ | . . . | $V_3$ | . . . |

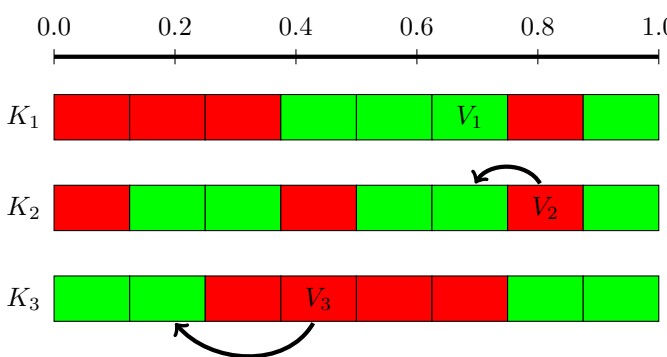

Figure 1: Illustrative example of Algorithm 1 on a tabular dataset with 3 rows and 4 columns. This structure corresponds to 2 pairs of $(key, value)$ columns. In the first row, the element $V_1$ is already in a 'green' bin. Meanwhile, the other elements, $V_2$ and $V_3$, have to be moved from 'red' interval to a nearby 'green' interval.

have proposed watermarking for tabular data, they often fail to address how their watermark performs under these innocuous attacks masked as preprocessing steps.

This paper focuses on establishing a theoretical framework for watermarking tabular datasets. Our approach leverages the structure of the feature space to form pairs of $(key, value)$ columns for a more fine-grained watermark embedding. The elements in the *key* columns are divided into consecutive intervals, whose centers determine the seed to generate random 'red' and 'green' intervals for the corresponding *value* columns. We embed the watermark into the data by promoting all elements in the *value* columns to fall into these green intervals. For an illustrative example of this watermarking process on a stylized dataset, see Figure 1. During the detection phase, we employ a hypothesis test to examine the characteristics of the empirical data distribution. Furthermore, we provide theoretical and empirical analysis of the watermark's robustness to various oblivious attacks on synthetic and real-world datasets. Finally, we provide the first set of theoretical results on the query complexity for an adversary to learn the watermarking scheme. We summarize our main contributions as follows:

- In Section 4, we propose a fine-grained watermarking scheme for tabular data with strong statistical guarantees and desirable properties: fidelity, detectability, robustness, and security. To our knowledge, this is the first work that leverages the feature space structure to watermark tabular datasets.

- To enhance robustness and ensure high-quality data for downstream tasks, we employ a pairing subroutine based on the feature importance. Our proposed pairing scheme retains twice as many essential columns under the feature selection attack as the naive approach.

- In Section 7, we give an upper bound on the query complexity required by an adversary to learn our watermarking scheme under two querying schema: table-like queries and row-like queries. This is the first theoretical result on statistical learning of tabular watermarking techniques.

- In Section 8, we demonstrate our proposed watermarking method's high fidelity, detectability, robustness and security on a set of synthetic and real-world datasets.

## 2  Related Work

There exists a long line of research studying watermarking schemes for tabular data. Agrawal & Kiernan (2002) is the first work to tackle this problem setting, where the watermark was embedded in the least significant bit of some cells, i.e., setting them to be either 0 or 1 based on a hash value computed using primary and private keys. Subsequently, Xiao et al. (2007); Hamadou et al. (2011) proposed an improved watermarking scheme by embedding multiple bits. Another approach is to embed the watermark into the data statistics. Notably, Sion et al. (2003) partitioned the rows into different subsets and embedded the watermark

by modifying the subset-related statistics. This approach is then improved upon by Shehab et al. (2007) to protect against insertion and deletion attacks by optimizing the partitioning algorithm and using hash values based on primary and private keys. To minimize data distortion, the authors modeled the watermark as an optimization problem solved using genetic algorithms and pattern search methods. However, this approach is strictly limited by the requirement for data distribution and the reliance on primary keys for partitioning algorithms.

Inspired by the recent watermarking techniques in large language models (Kamaruddin et al., 2018; Aaronson, 2023; Kuditipudi et al., 2024; Kirchenbauer et al., 2023), there are two primary approaches for watermarking generative tabular data: post-process watermarks and generation-time watermarks. Particularly, the generation-time tabular watermark approach in (Zhu et al., 2025) is closely related to existing watermarking techniques with diffusion models in images (Wen et al., 2023; Yang et al., 2024). While this watermark technique demonstrate both theoretical and empirical effectiveness and robustness, it heavily depends on DDIM (Song et al., 2022), which may limits its practicality. In a separate work, Fang et al. (2025) proposes to embed the watermark at the end of the sampling stage by tournament sampling. Specifically, the watermark mechanism selects multiple rows with high scores for a random function seeded by special columns in the generated dataset to be watermarked samples. However, this generation-time tabular watermark approach depends on the watermark owner having access to the sampling process, which is not always the case in practice.

In this work, we focus on post-process watermark that is model-agnostic, where the watermark is embedded in the data after it has been generated. Prior work in post-process watermark (He et al., 2024; Zheng et al., 2024) has made advances in watermarking generative tabular data by splitting the value range into red and green intervals. He et al. (2024) proposed a watermarking scheme through data binning, ensuring that all elements in the original data are close to a green interval. In conjunction with a statistical hypothesis test for detection, this embedding method allows the authors to protect the watermark from additive noise attack. However, the authors assumed that both the tabular dataset and the additive noise elements are sampled from a continuous distribution, which does not account for attacks such as feature selection or truncation. Zheng et al. (2024) instead only embedded the watermark in the prediction target feature. While the authors showed that this watermarking technique can handle several attacks and categorical features, their results primarily focused on watermarking one feature using a random seed, which is often insufficient in practice. In contrast, our technique guarantees that half of the dataset is watermarked with the seed chosen based on the data in *key* columns. As a result, our watermark approach is more robust to dataset-level manipulations such as feature subset attacks, especially when the target feature for downstream tasks is not known in advance. Moreover, we observe an interesting tradeoff between robustness and detection cost among these prior work and our approach: with the same level of fidelity, He et al. (2024)'s watermark is the easiest to detect but is less robust to oblivious and adversarial attacks compared to Zheng et al. (2024) and our watermarks. On the other hand, we focus on enhancing the robustness of the watermark against potential attacks at the cost of harder detection due to the need to recover the exact $(key, value)$ feature pairs. Finally, Zheng et al. (2024) offers a watermark algorithm that works for both continuous and categorical features with mildly higher detection cost and robustness compared to He et al. (2024). For a high level summary of the comparison between our method and these two works, see Table 1.

Particularly, both the prior works (He et al., 2024; Zheng et al., 2024) and our proposed approach maintain high data fidelity, measured by comparing the utility of using the watermarked dataset compared to the original unwatermarked dataset on downstream machine learning tasks. While all three approaches use a similar hypothesis testing procedure to detect the watermark, He et al. (2024)'s watermark is the easiest to detect. The additional cost of watermark detection in Zheng et al. (2024) and our schemes are due to the selection of the watermarked features. At detection time, the third-party need to identify either the target watermarked feature (for Zheng et al. (2024)'s watermark) or all (key, value) feature pairs (for our approach). Compared to He et al. (2024), our approach is more robust to a simple spoofing attack (details in Section 8.2), where an adversary attempts to create harmful content with a target watermark embedded (Jovanović et al., 2024). On the other hand, while Zheng et al. (2024) provided a comprehensive study of potential attacks against their watermark, their approach is less robust to the feature selection attack compared to ours. This difference stems from how the watermarked feature is chosen in their watermark. Since Zheng et al. (2024)

mainly watermarks only the target feature (with possible extensions to multiple features), if an oblivious data scientist decides to drop this watermarked feature then the watermark is effectively erased. Meanwhile, we specifically design the $(key, value)$ PAIR routine (Section 6.1) to preserve the watermark even when a large number of columns is dropped. For security attacks on the watermark, we provide the first upper bound on query complexity required by an adversary to learn the watermark scheme. Under our analysis Section 7, this upper bound is the same for all three tabular watermarking approaches.

| | He et al. (2024) | Zheng et al. (2024) | **Ours** |
|---|---|---|---|
| Data Type | Numerical | Numerical, Categorical | Numerical |
| Fidelity | $1/b$ | $\min\left\{1/2b\left(\log\left(m/\delta\right)-1\right),1\right\}^1$ | $\min\left\{1/2b\left(\log\left(mn/\delta\right)-1\right),1\right\}$ |
| Detection Test | $\chi^2$-test | One Proportion $z$-test | One Proportion $z$-test with Bonferroni correction |
| Robustness | Additive noise | Additive Noise, Insertion, Deletion, Shuffle, Row Subset, Invertibility | Additive Noise, Truncation, Feature Selection |
| Spoofing | No | Yes | Yes |
| Learning | $\tilde{O}(mnb)$ | $\tilde{O}(mnb)$ | $\tilde{O}(mnb)$ |

Table 1: A comparison of the data type and results from (He et al., 2024; Zheng et al., 2024) and our technique for post-process watermarking generative tabular data. The fidelity measures the $L_\infty$ distance between the original and watermarked datasets. For spoofing attack, we list whether the watermark is robust against a simple spoofing procedure (described in Section 8.2), where an adversary attempts to generate harmful content that carries the target watermark. The statistical learning upper bounds represent the query complexity required by an adversary to learn the watermark scheme.

## 3 Problem Formulation

**Notation.** For $n \in \mathbb{N}^+$, we write $[n]$ to denote $\{1, \cdots, n\}$. For a matrix $\mathbf{X} \in \mathbb{R}^{m \times 2n}$, we denote the $L$-infinity norm of $\mathbf{X}$ as $\|\mathbf{X}\|_\infty = \max_{i \in [m], j \in [2n]} |\mathbf{X}_{i,j}|$. For an interval $g = [a, b]$, we denote the center of $g$ as $\text{center}(g) = (a+b)/2$.

In this paper, we consider an original tabular dataset $\mathbf{X} \in [0,1]^{m \times 2n}$ with each column containing $m$ i.i.d data points from a (possibly unknown) distribution $F_i, i \in [2n]$ with continuous probability density function $f_i$. Our main objective is to generate a watermarked version of this data, denoted $\mathbf{X}_w$, that achieves the following properties:

(i) **Fidelity**: the watermarked dataset $\mathbf{X}_w$ is close to the original data set $\mathbf{X}$ to maintain high fidelity, measured through $L_\infty$ distance and Wasserstein distance;

(ii) **Detectability**: the watermarked dataset $\mathbf{X}_w$ can be reliably identified through the one-proportion $z$-test using only few row samples;

(iii) **Robustness**: the watermarked dataset $\mathbf{X}_w$ achieves desirable robustness against multiple methods of oblivious 'attack' commonly observed in data science;

(iv) **Security**: it is difficult for an adversary to learn the underlying watermarking scheme without considerable computational efforts.

Note that in our analysis, we only focus on embedding the watermark in the continuous features of a tabular dataset. While a real-world tabular dataset may contain many categorical features, embedding the watermark

---

[1]Zheng et al. (2024) did not provide theoretical analysis of the fidelity. This bound comes from our analysis with a slightly different union bound event.

---

**Algorithm 1:** Pairwise Tabular Watermarking

---

**Input:** Tabular dataset $\mathbf{X} \in \mathbb{R}^{m \times 2n}$. Number of bins $b \in \mathbb{N}^+$. Pairing subroutine PAIR.
**Output:** Watermarked dataset $\mathbf{X}_w$.
 1: Divide the columns into $n$ pairs of columns labeled $(key, value)$ using PAIR.
 2: Divide the *key* columns into bins of equal width $1/b$ to form consecutive intervals denoted $\{I_j\}_{j \in [b]}$, where $I_j = [j - 1/b, j/b]$.
 3: **for** each *key* column **do**
 4:     Use the center of the bins to compute the hash and seed a random number generator.
 5:     Randomly generate red and green intervals, ensuring equal number of each type. Denote the set of green intervals as $G$.
 6:     **for** each element $x$ in the paired *value* column **do**
 7:         Identify the nearest green interval as $g = \arg\min_{g \in G} |x - \mathrm{center}(g)|$.
 8:         **if** $x \notin g$ **then**
 9:             Replace $x$ with $x_w$ uniformly sampled from $g$.
10:         **else**
11:             Leave $x$ as is.

---

in these features through a small perturbation of its value may cause significant changes in the meaning for the entire sample (row). Given a dataset $\mathbf{X}$ with both categorical and continuous features, we can run our algorithm on a smaller dataset $\mathbf{X}' \subseteq \mathbf{X}$ with same number of rows and only continuous features. After the watermark has been embedded in $\mathbf{X}'$ to get $\mathbf{X}'_w$, we reconstruct a watermarked version of the original dataset $\mathbf{X}$ by replacing the continuous features of $\mathbf{X}$ with its watermarked versions from $\mathbf{X}'_w$. We leave the study of watermarking categorical features in tabular datasets for future work.

## 4 Pairwise Tabular Data Watermark

In this section, we provide the details of the watermarking algorithm. At a high level, we embed the watermark into a tabular dataset $\mathbf{X}$ by leveraging the pairwise structure of its feature space. Particularly, we first partition the feature space into pairs of $(key, value)$ columns using a subroutine PAIR. Detail of this subroutine will be discussed in the later section. Then, we finely divide the range of elements in each *key* column into bins of size $1/b$ to form $b$ consecutive intervals. The center of the bins for each *key* column is then used to compute a hash, which becomes the seed for a random number generator. This random number generator then randomly generates red and green intervals for the corresponding *value* column, where each interval is of size $1/b$. For simplicity, in the remaining analysis, we assume that half of the intervals is red (and the other half is green). Finally, we embed the watermark in elements of this *value* column by ensuring that they are always in a nearby green interval with minimal distortion. This process is repeated until all *value* columns are watermarked. We formally describe our proposed watermarking method in Algorithm 1 below.

### 4.1 Fidelity of Pairwise Tabular Data Watermark

First, we show that the watermarked dataset $\mathbf{X}_w$ maintains high fidelity, i.e., $\mathbf{X}_w$ is close to the $\mathbf{X}$ in $L_\infty$ distance. Since the red and green intervals are randomly assigned, our bound on the $L_\infty$ distance holds with high probability. Intuitively, this bound depends on the distance to the nearest green interval: the probability that a search radius contains a green interval grows with the number of adjacent bins included in the search.

**Theorem 4.1** (Fidelity). *Let $\mathbf{X} \in [0, 1]^{m \times 2n}$ be a tabular dataset, and $\mathbf{X}_w$ is its watermarked version from Algorithm 1. With probability at least $1 - \delta$ for $\delta \in (0, 1)$, the $L_\infty$ distance between $\mathbf{X}$ and $\mathbf{X}_w$ is upper bounded by:*

$$\|\mathbf{X} - \mathbf{X}_w\|_\infty \le \min\left\{\frac{1}{2b}\left(\log_2\left(\frac{mn}{\delta}\right) - 1\right), 1\right\} \tag{1}$$

Theorem 4.1 gives rise to a natural corollary that upper bounds the Wasserstein distance, i.e., the distance between the empirical distributions of $\mathbf{X}$ and $\mathbf{X}_w$. Together, Theorem 4.1 and Corollary 4.2 show that

in expectation, the watermarked dataset $\mathbf{X}_w$ is close to the original dataset $\mathbf{X}$. Thus, downstream tasks operated on $\mathbf{X}_w$ instead of $\mathbf{X}$ would only induce additional error in the order of $1/b$ with high probability. We empirically show the impact of this additional error for several synthetic and real-world datasets in Section 8.

**Corollary 4.2** (Wasserstein distance)**.** *Let* $F_{\mathbf{X}} = \sum_{j=1}^{m} \frac{1}{m} \delta_{\mathbf{X}[j,:]}$ *be the empirical distribution built on* $\mathbf{X} \in [0,1]^{m \times 2n}$, *and* $F_{\mathbf{X}_w} = \sum_{j=1}^{m} \frac{1}{m} \delta_{\mathbf{X}_w[j,:]}$ *be the empirical distribution built on* $\mathbf{X}_w$. *With probability at least* $1 - \delta \in (0,1)$, *the k-Wasserstein distance is upper bounded by*

$$\mathcal{W}_k(F_{\mathbf{X}}, F_{\mathbf{X}_w}) \leq \frac{\sqrt{n/2}}{b} \cdot \log_2\left(\frac{mn}{\delta}\right) \tag{2}$$

## 5 Detection of Pairwise Tabular Data Watermark

In this section, we provide the details of the watermark detection protocol. Informally, the detection protocol employs standard statistical measures to determine whether a dataset is watermarked with minimal knowledge assumptions. Particularly, we introduce the following lemma that shows, with an increasing number of bins, the probability of any element in a *value* column belonging to a green interval approaches $1/2$. That is, without running the watermarking algorithm 1, we have a baseline for the expected number of elements in green intervals for each *value* column.

**Lemma 5.1.** *Let F be the probability distribution of the data x which is to be watermarked (a single element of a value column of the input table) with support in* $[0,1]$. *Define G as the union of intervals which are labeled green. Then, we have* $\Pr[x \in G] = 1/2$ *where the probability is over* $x \sim F$ *and over the choice of the labels of the intervals (red/green).*

We formalize the process of detecting our watermark through a hypothesis test. Intuitively, the result of Lemma 5.1 implies that, for any *value* column, the probability of an element being in a green list interval is approximately $1/2$. While this convergence is agnostic to how the green intervals are chosen, it is non-trivial for a data-provider to detect the watermark due to the pairwise structure of Algorithm 1. Particularly, if the data-provider has knowledge of the *value* columns and the hash function, they still need to individually check which *key* column corresponds to the selected *value* column. In the worst case, all $n$ *key* columns must be checked for each *value* column before the data-provider can confidently claim that the dataset is not watermarked. With this knowledge, we formulate the hypothesis test as follows:

**Hypothesis test**

$H_0$ : Dataset $X$ is not watermarked $\qquad\qquad$ $H_{0,i}$ : The $i$-th *value* column is not watermarked

$H_1$ : Dataset $X$ is watermarked

That is, when the null hypothesis holds, all of the individual null hypotheses for the $i$-th value column must hold simultaneously. Thus, the data-owner who wants to detect the watermark for a dataset $\mathbf{X}$ would need to perform the hypothesis test for each *value* column individually. If the goal is to reject the null hypothesis $H_0$ when the $p$-value is less than a predetermined significant threshold $\alpha$ (typically 0.05 to represent 5% risk of incorrectly rejecting the null hypothesis), then the data-provider would check if the $p$-value for each individual null hypothesis $H_{0,i}$ is lower than $\alpha/n^2$ (after accounting for the family-wise error rate using Bonferroni (Bonferroni, 1936)).[2]

Let $T_i$ denote the number of elements in the $i$-th *value* column that falls into a green interval. Then, under the individual null hypothesis $H_{0,i}$, we know that $T_i \sim B(m, 1/2)$ for large number of rows $m$. Using the Central Limit Theorem, we have $2\sqrt{m}\left(\frac{T_i}{m} - \frac{1}{2}\right) \to \mathcal{N}(0,1)$. Hence, the statistic for a one-proportion $z$-test is

$$z = 2\sqrt{m}\left(\frac{T_i}{m} - \frac{1}{2}\right) \tag{3}$$

---

[2]With knowledge of the feature importance (detail in Section 6), the data-owner can choose adaptive significant level $\alpha$ for each individual hypothesis test. Informally, we put more weight on pairs of (*key*, *value*) columns that are closer together in their feature importance while ensuring that all significant levels sum up to the desired $\alpha$ threshold.

For a given pair of ($key, value$) columns, the data-owner can calculate the corresponding $z$-score by counting the number of elements in $value$ column that are in green intervals. Since we are performing multiple hypothesis tests simultaneously if the dataset has 10 pairs of columns and the chosen significant level $\alpha = 0.05$, then the individual threshold for each column is $\alpha_i = 0.0005$. The data-owner can look up the corresponding threshold for the $z$-score to reject each individual null hypothesis. If the calculated $z$-score exceeds the threshold, then the data-provider can reject the null hypothesis and claim that this $value$ column is watermarked. On the other hand, if the $z$-score is below the threshold, then the data-owner cannot conclude whether this $value$ column is watermarked or not until they have checked all possible $key$ columns.

## 6 Robustness of Pairwise Tabular Data Watermark

In this section, we examine the robustness of the watermarked dataset when they are subjected to different 'attacks' commonly seen in data science. We assume that the attacker has no knowledge of the ($key, value$) pairing scheme and, consequently, has no knowledge of the green intervals. We focus on two types of attacks: feature extraction and truncation, which are common preprocess steps before the dataset can be used for a downstream task.

### 6.1 Robustness to Feature Selection

Given a watermarked dataset $\mathbf{X}_w$ and a downstream task, a data scientist may want to preprocess the data by dropping irrelevant features from $\mathbf{X}_w$. Formally, we make the following assumption on how to perform feature selection:

**Assumption 6.1.** *Given a dataset $\mathbf{X} \in [0,1]^{m \times 2n}$ with features $\mathbf{X}_1, \cdots, \mathbf{X}_{2n}$, one can perform feature selection according to a known feature importance order with regard to the downstream task. Then, the truncated dataset is of size $m \times k$ for $k \leq 2n$, where only the top-k features with the highest importance are kept from the original dataset.*

Algorithm 1 takes a black-box pairing subroutine PAIR as an input to determine the set of ($key, value$) columns. In the following analysis, we consider two feature pairing schemes: (i) *uniform*: features are paired uniformly at random, or (ii) *feature importance*: features are paired according to the feature importance ordering, where features with similar importance are paired. Without loss of generality, we assume that the columns of the original dataset are ordered in descending order of feature importance. Note that this reordering of features does not affect the uniform pairing scheme and only serves to simplify notations in our analysis. Formally, given two columns $\mathbf{X}_i$ and $\mathbf{X}_j$, we define the probability of $(\mathbf{X}_i, \mathbf{X}_j)$ being a ($key, value$) pair as proportional to the inverse of the distance between their indices.

$$\Pr[(\mathbf{X}_i, \mathbf{X}_j) \text{ is pair}] = \frac{\frac{1}{|i-j|}}{\sum_{\ell \in [2n], \ell \neq i} \frac{1}{|i-\ell|}} \tag{4}$$

In the following theorem, we show that feature importance pairing will preserve more pairs of columns after the feature selection attack compared to uniform pairing.

**Theorem 6.2.** *Given a watermarked dataset $\mathbf{X}_w$ and a data scientist attacking $\mathbf{X}_w$ with feature selection as in Assumption 6.1. Then, the number of preserved column pairs under feature importance pairing is at least twice as many as that under uniformly random pairing.*

Theorem 6.2 implies that, under the feature importance pairing scheme, the truncated dataset would retain more valuable information. In Section 8, we empirically show how this theoretical guarantee translates to improved utility in the downstream task for various datasets.

### 6.2 Robustness to Truncation

In addition to feature selection, the data scientist can also 'attack' the watermarked dataset by directly modifying elements in the dataset. In particular, we are interested in 'truncation' attack, where the data

scientist reduces the number of digits after the decimal point of all elements in the dataset. Formally, let truncate : $\mathbb{R} \to \mathbb{R}$ be the truncation function defined as:

$$x_{\mathrm{tr}} = \mathrm{truncate}(x, p) = \frac{\lfloor 10^p \cdot x \rfloor}{10^p} \tag{5}$$

That is, for all $x \in \mathbf{X}_w$, the data scientist truncates the digits in the mantissa of $x$ to $x_{\mathrm{tr}} \in \mathbb{R}$ with $p$ digits in the mantissa. For example, with $x = 0.369$ and $p = 2$, the data scientist will truncate $x$ to get $x_{\mathrm{tr}} = 0.36$. In the following analysis, we focus on the case where $p = 2$, i.e., all values are truncated to 2 decimal places. Extension to more digits in the mantissa follows the same analysis.

First, we determine how this truncation operation influence the distribution of watermarked elements in green intervals. When a watermarked element $x$ in a *value* column is truncated to $x_{\mathrm{tr}}$, it can fall out of the original green interval if the bins $[0, 1/b], \cdots, [b-1/b, 1]$ and the hundredth grid points $\{0, 0.01, 0.02, \cdots, 0.99, 1\}$ are not perfectly aligned. To illustrate this phenomenon, we presented stylized example where Algorithm 1 uses $b = 150$ bins for its watermarking procedure. Then, in the second bin $I_2 = [1/150, 2/150]$, any element $x \in [1/150, 0.01)$ will be truncated to $x_{\mathrm{tr}} = 0.0 \in I_1$. If $I_1$ is chosen to be a red interval by the random number generator in Algorithm 1, then the truncation operation has successfully moved elements out of the green intervals. In the following theorem, we show the probability of successful truncation attack as a function of the bin width.

**Theorem 6.3.** *Given a watermarked element $x \in I_j = [j-1/b, j/b]$ and the truncation function defined in Equation* (5). *Then, the probability that the truncated element $x_{\mathrm{tr}}$ falls out of its original green interval is*

$$\Pr[x_{\mathrm{tr}} \notin I_j] = \frac{(b-1)^{100} + b^{99}(c \cdot b - j + 1)}{b^{100}}$$

*where $c \in \{0.00, 0.01, \cdots, 1.00\}$ is the left grid point in $I_j$.*

Intuitively, with larger bin size $1/b$, the probability that our proposed watermarking scheme can withstand truncation attack increases as the truncated elements are more likely to fall into the same bins as the original elements. On the other hand, when the bins are more fine-grained, truncation would almost surely move the watermarked data outside of the original intervals. This result presents an interesting trade-off between smaller bin width for higher fidelity (see Theorem 4.1) and bigger bin width for better robustness, which has not been studied in prior work on watermarking tabular data. With this insight, we can choose the bin width to be the same as the truncation grid size, i.e., $1/b = 1/10^p$ or $b = 10^p$ to ensure high fidelity and robustness.

## 7 Decoding watermark

It is important to understand how many queries are required by an adversary to learn a watermarking scheme. Specifically, this would allow us to bound the total number of queries we can allow one single user before we run the risk of giving out our algorithms. Although such a result has not been studied in prior work (He et al., 2024), we believe it would showcase the power of any watermarking scheme. To this end, we demonstrate that the number of queries an adversary would require to learn the watermarking algorithm presented in our work (Algorithm 1) is $\tilde{O}(mnb)$, where $b$ is the number of bins used to embed the watermark in Algorithm 1, and the adversary uses a query table of size $m \times n$. We informally state this result below, and defer the formal statement and its proof until Appendix E.

**Theorem** (E.2, Informal)**.** *The watermarking scheme in Algorithm 1 can be learned up to a small error $\epsilon \in (0, 1)$ with probability at least $1 - \delta \in (0, 1)$ by making $O\left(\frac{mnb \log(mnb) \log(1/\epsilon) + \log(1/\delta)}{\epsilon}\right)$ queries to the detector using tables of size $m \times n$.*

In practice, any watermark provider using Algorithm 1 to watermark their data can artificially limit the total number of queries by order or magnitudes less than the query complexity of Theorem E.2 to avoid divulging the watermarking scheme. We also observe that the query complexity of learning Algorithm 1 is exactly same as the query complexity of learning the algorithm of He et al. (2024). This is an artifact of how we derive our bounds (via VC dimension). Beyond table like queries, we also study the case when queries are of the form of rows. The formal theorem statement and an analysis for the case where queries are of the form of a row can be found in Appendix E.3.

# 8 Experiments

In this section, we empirically evaluate the fidelity and robustness of our watermarking algorithm on a set of synthetic and real-world datasets.

## 8.1 Synthetic Tabular Data

We begin our evaluation using synthetic data to validate the properties of our algorithm, including fidelity, robustness, and performance sensitivity to selected parameters.

### 8.1.1 Experimental Detail

We generate a dataset of size $2000 \times 2$ using the standard Gaussian distribution. One column is designated as the seed column, and the other is watermarked. Figures 2a and 2b contain KDE plots showing a minimal difference between the distribution before and after watermarking.

In the second experiment, we generate multiple datasets, each containing 50 columns and varying numbers of rows from 20 to 100, to validate the effect that the row count has on the maximum achievable $z$-score. We repeat this process 5 times and take the average $z$-score. In Figure 2c, we find that as the number of rows in the dataset increases, so does the maximum possible $z$-score. This means that given a choice of $z$-score threshold, i) there is an increasing minimum number of total rows that the dataset must contain to achieve that score, and ii) as the number of rows increases, so does the number of rows that an attacker must sufficiently alter to break the watermark.

Finally, we consider the *fidelity vs. robustness* trade-off that the *bin size* parameter poses. We generate a dataset of size $2000 \times 2$ using the standard Gaussian distribution. We watermark one column, using the other for seeding, and vary the *bin size* between $10^{-4}$ and $10^{-1}$. The average mean squared error across 5 runs between the original data and the watermarked data for each *bin size* is shown in Figure 2d. As expected, greater fidelity is maintained with smaller bins since adjusting the data to fall into the nearest green bin requires smaller perturbations. Figure 2e displays the accompanying susceptibility to noise that comes with smaller bins. Across 5 runs, we add zero-mean Gaussian noise with standard deviation varying from $10^{-3}$ to $10^{-1}$ to the watermarked data and measure the effect on the $z$-score. In each case, we find that smaller bins result in lower scores.

### 8.1.2 Classification

To validate the effectiveness of our feature importance based pairing scheme, we generate a multi-class classification dataset of size $75 \times 75$ using scikit-learn, setting the number of classes to 5 and the number of informative features to 37. We create a set of column pairs using two different schemes: i) *uniform*: features are paired uniformly at random and ii) *feature importance*: sampling columns by treating their feature importance according to a Random Forest classifier as probabilities, pairing columns that are sampled one after the other. Using these two pairing schemes we create two watermarked datasets by watermarking 12 of the 37 available pairs, respectively.

For each of the watermarked datasets, we train a Random Forest classifier and use the resulting feature importance to drop subsets of columns varying in size from 20% to 80%. The metric of interest in this experiment is the percentage of pairs retained after column dropping. Both columns in a pair must still remain in the dataset to be counted as a preserved pair. This entire process is repeated 5 times, and the averaged results are shown in Figure 2f. The results show a significant increase in preserved pairs when the pairs are created using the feature importance scheme.

## 8.2 Spoofing

We introduce a novel method (Algorithm 2) to spoof and test the robustness of our algorithm by comparing it to the previous work of He et al. (2024). This spoofing algorithm is designed to subtly alter a dataset by modifying the decimal points of its element, effectively inserting a watermark to the dataset. The process

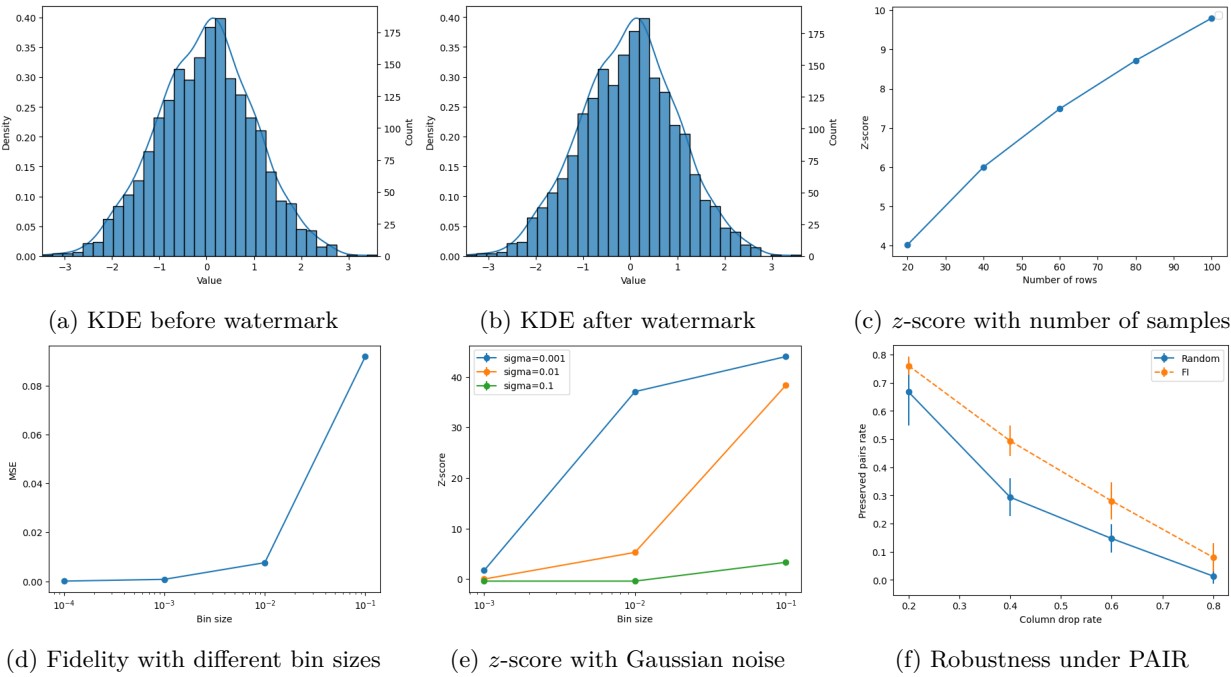

Figure 2: (a) KDE plot of the Gaussian data before watermarking. (b) KDE plot of the Gaussian data after watermarking. (c) The maximum possible $z$-score increases with the number of rows in the dataset. (d) Smaller bin sizes result in higher fidelity (lower MSE) between the original data and the watermarked data. (e) Smaller bin sizes are more susceptible to noise, resulting in lower $z$-scores. Sigma corresponds to the standard deviation of the applied zero-mean Gaussian noise. (f) Pairing columns according to feature importance increases the pair preservation rate when the least important features are dropped. 'Random' indicates random column pairings while 'FI' indicates column pairings biased toward similar feature importances.

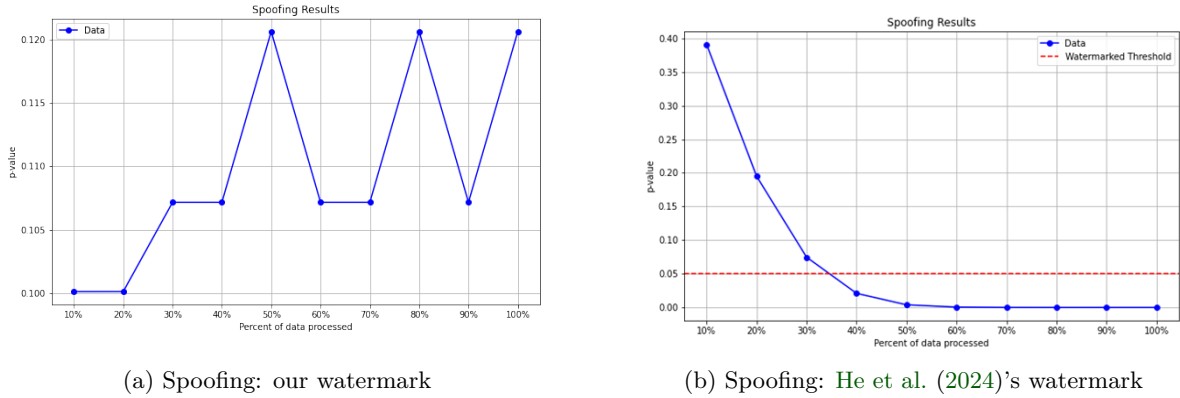

Figure 3: (a) Plot of our replacement spoofing algorithm on our watermark approach. Our approach is highly robust as the spoofing algorithm as well as the percent of data processed has no effect on our dataset. (b) Plot of our replacement spoofing algorithm on He et al. (2024)'s watermark approach. As the percentage of data processed increases, the $p$-value decreases. As the percent of data processed increases near 40%, the $p$-value of He et al. (2024)'s watermark is already below 0.05 and the watermark can be spoofed using Algorithm 2.

begins by computing the integers and fractional parts of a synthetic dataset and a reference watermarked dataset. For the experiment shown below Figure 3b, we generated a synthetic dataset of the original unwatermarked dataset to act as our target. For each element in the synthetic dataset, the algorithm identifies

the closest fractional part from the watermarked dataset to the current element. If the element is non-negative, the fractional part is added to the integer part. This method ensures that the watermark is embedded in a way that is minimally invasive, preserving the fidelity, detectability, overall structure and integrity of the original data while introducing a subtle, detectable pattern. The experimental results are in Figure 3.

Using Algorithm 2, we can successfully spoof the watermark scheme by prior work (He et al., 2024) after replacing about 40% of the synthetic dataset. That is, the spoofed dataset generated from Algorithm 2 will be classified as 'watermarked' by the He et al. (2024)'s detector as the $p$-value drops below 0.05. On the other hand, our watermark approach (Algorithm 1) cannot be spoofed by using Algorithm 2 at all. Even after all data points in the synthetic dataset have been processed, the $p$-value from our detection hypothesis test (Section 5) still remains above 0.05. This result shows that our watermark approach is more robust to simple spoofing attack compared to prior work of He et al. (2024) by leveraging the structure of the tabular dataset.

---

**Algorithm 2:** Fractional Replacement

---

**Input:** Synthetic dataset $\mathbf{S} \in \mathbb{R}^{m \times n}$, Watermarked dataset $\mathbf{W} \in \mathbb{R}^{m \times n}$.
**Output:** Modified dataset $\mathbf{X}'$.
 1: Compute the fractional parts of $\mathbf{S}$ to form $\mathbf{S\_fractions}$.
 2: Compute the fractional parts of $\mathbf{W}$ to form $\mathbf{W\_fractions}$.
 3: **for** each element $x_{ij}$ in $\mathbf{S}$ **do**
 4:     Extract the integer part $\lfloor x_{ij} \rfloor$.
 5:     Compute the fractional part frac\_part $= |x_{ij} - \lfloor x_{ij} \rfloor|$.
 6:     Find the closest fractional part closest\_frac $= \arg\min_{f \in \mathbf{S\_fractions} \cup \mathbf{W\_fractions}} |f - \text{frac\_part}|$.
 7:     **if** $x_{ij} < 0$ **then**
 8:         Replace $x_{ij}$ with $\lfloor x_{ij} \rfloor - \text{closest\_frac}$.
 9:     **else**
10:         Replace $x_{ij}$ with $\lfloor x_{ij} \rfloor + \text{closest\_frac}$.

---

### 8.3 Generative Tabular Data

In addition to synthetic data, we evaluate the theoretical guarantees of our watermark approach on a set of real-world datasets.

#### 8.3.1 Datasets and Generators

Similar to He et al. (2024), we evaluate our proposed watermarking technique using two real-world datasets of various sizes and distributions: Wilt (Johnson, 2014), and California Housing Prices.

***Wilt.*** This dataset is a public dataset that is part of the UCI Machine Learning Repository. It contains data from a remote sensing study that involved detecting diseased trees in satellite imagery. The data set consists of image segments generated by segmenting the pan-sharpened image. The segments contain spectral information from the Quickbird multispectral image bands and texture information from the panchromatic (Pan) image band[3]. This dataset includes 4,889 records and 6 attributes. The attributes are a mixture of numerical and categorical data types. The data set has a binary target label, which indicates whether a tree is wilted or healthy. Therefore, the dataset has a classification task, which is to classify the tree samples as either diseased or healthy.

***California Housing Prices.*** The dataset was collected from the 1990 U.S. Census and includes various socio-economic and geographical features that are believed to influence housing prices in a given California district. It contains 20,640 records and 10 attributes, each of which represents data about homes in the district. Similar to the previous dataset, the attributes are a mixture of continuous and categorical data types. The dataset has a multi-target label, which indicates the proximity of each house to the ocean, making it a multi-classification problem.

---

[3]Data available on the UCI platform at https://archive.ics.uci.edu/dataset/285/wilt

For each dataset, we generate corresponding synthetic datasets using neural network-based methods Park et al. (2018) and statistical-based generative methods Li et al. (2020). Each of these methods for synthetic data generation possesses distinctive capabilities and features. For this paper, we employ CTGAN (Xu et al., 2019), Gaussian Copula (Masarotto & Varin, 2012), and TVAE (Xu et al., 2019) which represent GAN-based (Goodfellow et al., 2014), copula-based (Patki et al., 2016), and VAE-based generators (Kingma & Welling, 2013) respectively to generate tabular data.

### 8.3.2 Utility

| Dataset | Method | Not WM | Watermarked (WM) | | | | | |
|---------|--------|--------|------|------------------|-------------------------|--------|-------------------------|--------|
| | | | WM | WM and Truncated | WM and 20% cols drop | | WM and 40% cols drop | |
| | | | | | FI | Random | FI | Random |
| California | CTGAN | 0.373 | 0.371 | 0.368 | 0.301 | 0.242 | 0.203 | 0.256 |
| | Copula | 0.370 | 0.376 | 0.376 | 0.347 | 0.332 | 0.31 | 0.30 |
| | TVAE | 0.797 | 0.799 | 0.798 | 0.448 | 0.407 | 0.385 | 0.365 |
| Wilt | CTGAN | 0.731 | 0.733 | 0.733 | 0.575 | 0.574 | 0.563 | 0.563 |
| | Copula | 0.99 | 0.996 | 0.996 | 0.995 | 0.994 | 0.993 | 0.993 |
| | TVAE | 0.989 | 0.989 | 0.989 | 0.965 | 0.977 | 0.972 | 0.803 |

Table 2: Accuracy of the downstream models under various attacks to the watermarked datasets. In particular, we provide accuracy for the original dataset and its watermarked counterpart. Additionally, we consider truncation as well as the column dropping separately that are typical preprocessing steps in a machine learning pipeline. We note that the effect of our watermarking technique is negligible in terms of accuracy in all cases while maintaining high detectability.

For evaluating utility, we focus on machine learning (ML) efficiency (Xu et al., 2019). In more detail, ML efficiency quantifies the performance of classification or regression models that are trained on synthetic data and evaluated on the real test set. In our experiment, we evaluate ML efficiency with respect to the XGBoost classifier (Chen & Guestrin, 2016) for classification tasks, which are then evaluated on real testing sets. Classification performances are evaluated by the accuracy score. To explore the performance of our tabular watermark on real-world data, we sample from each generative model a generated dataset with the size of a real training set. For each setup, we create 5 watermarked training sets to measure the accuracy score. For reproducibility, we set a specific random seed to ensure that the data deformation and model training effects are repeatable on a similar hardware and software stack. To eliminate the randomness of the results, the experimental outcomes are averaged over multiple runs from each watermarked training set.

We watermark each of the generated datasets using a *bin size* of $10^{-2}$ and thus only consider columns that contain floating point numbers with at least 2 decimal places. This choice follows from the practical consideration that watermarking with this bin size involves perturbing up to 2 decimal places, and watermarking any original columns that did not already contain values with this property may make it obvious to an outside party upon receiving the dataset that this specific section of the data has been manipulated.

We also consider two common data science preprocessing steps that downstream users of the watermarked datasets might conduct: truncation and dropping the least important columns. We aim to determine if the application of our watermark in conjunction with these operations For the former, we truncate to 2 decimal places. For the latter, we investigate dropping the lowest 20% and 40% of columns in each case, considering when the data is watermarked both with and without the feature importance-based pairing scheme.

We find that the effect of our watermarking technique is negligible in terms of accuracy in all cases while maintaining high detectability as seen in Table 2.

# 9 Discussion and Future Work

In this work, we provided a novel robust watermarking scheme for tabular numerical datasets. Our watermarking method partitions the feature space into pairs of ($key, value$) columns using knowledge of the feature importance in the downstream task. We use the center of the bins from each $key$ columns to generate randomized red and green intervals and watermark the $value$ columns by promoting its value to fall in green intervals. Compared to prior work in watermarking generative tabular data, our method is more robust to preprocess attacks such as feature selection and truncation at the cost of harder detection. There are a few open questions in the current work that are ripe for investigation:

- How do we include the categorical columns either in the key or the value of the watermarking process? Note that in the LLM setting, this was possible due to the richness of the vocabulary, which enabled replacing one token with a close enough token with a similar semantic meaning.

- Can we extend this framework to the LLM settings for tabular data generation? This extension embeds the watermark as part of the generation process (Venugopal et al., 2011) and the results in this paper will need to be adapted to the new setting. The samples generated by the LLM will be distorted due to the additional watermarking step, and recent work (Kuditipudi et al., 2024) mitigates it by inducing correlations with secret keys. A similar injection of undetectable watermarks has been studied by Christ et al. (2023). Adapting these to our settings would be of great interest.

- The current watermark scheme only considers a 'hard' watermark, where all elements in the $value$ columns are deterministically placed in the nearest green intervals. Can we extend the current analysis to allow for a 'soft' watermark scheme similar to the one described in Kirchenbauer et al. (2023), where we first promote the probability of being in green intervals, then sample from such distribution to generate elements for the $value$ columns?

- Our bounds for the query complexity to decode the query algorithm uses a specific embedding that allows us to apply VC dimensional bounds for learning the query function using neural networks with piecewise linear polynomial functions. It remains open whether one can derive a better query complexity bounds for learning the query function without resorting only to neural networks.

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

# A   Proof of Fidelity

**Proofs of Theorem 4.1**

*Proof.* Let $x$ be an element from the one of the value columns in the table $\mathbf{X}$, such that $x \in I_j$, where $I_j$ is the interval $\left[\frac{j-1}{b}, \frac{j}{b}\right]$ and the label of $I_j$ is red. Then, let $x_w$ be the corresponding watermarked value in $\mathbf{X}_w$. If $|x - x_w| > \frac{k}{b}$, then $x_w$ is either in one of the intervals $I_1, \ldots, I_{\frac{j-k-1}{b}}$ or in one of the intervals $I_{\frac{j+k+1}{b}}, \ldots, I_b$. By algorithm 1, we know that there are no green intervals between $I_{\frac{j-k-1}{b}}$ and $I_{\frac{j+k+1}{b}}$. Since we label each interval as red or green independently with probability $1/2$, the probability that there are no green intervals between $I_{\frac{j-k-1}{b}}$ and $I_{\frac{j+k+1}{b}}$ is at least $(\frac{1}{2})^{2k+1}$, where we use the fact that there are $2k + 2 + 1$ intervals between $I_{\frac{j-k-1}{b}}$ and $I_{\frac{j+k+1}{b}}$, and the first and last intervals in the sequence can be both green. Then we have that

$$\Pr(|x - x_w| > \frac{k}{b}) \le \frac{1}{2^{2k+1}}.$$

Now using the union bound over all $x$ in the value columns of $\mathbf{X}$, we have

$$\Pr\left( \|\mathbf{X} - \mathbf{X}_w\|_\infty > \frac{k}{b} \right) \le \frac{mn}{2^{2k+1}}.$$

Using $\frac{mn}{2^{2k+1}} \le \delta$ and solving for $k$ gives us

$$\Pr\left( \|\mathbf{X} - \mathbf{X}_w\|_\infty > \frac{1}{2b} \left( \log_2\left( \frac{mn}{\delta} \right) - 1 \right) \right) \le \delta.$$

Since $\|X - X_w\|_\infty \le 1$ always, we have with probability at least $1 - \delta$ for $\delta \in (0, 1)$,

$$\|\mathbf{X} - \mathbf{X}_w\|_\infty \le \min\left\{ \frac{1}{2b} \left( \log_2\left( \frac{mn}{\delta} \right) - 1 \right), 1 \right\}$$

$\square$

**Proof of Corollary 4.2**

*Proof.*

$$
\begin{aligned}
\mathcal{W}_k(F_{\mathbf{X}}, F_{\mathbf{X}_w}) &\le \left( \sum_{j=1}^m \frac{1}{m} \|\mathbf{X}[j,:] - \mathbf{X}_w[j,:]\|_2^k \right)^{1/k} \\
&\le \left( \sum_{j=1}^m \frac{1}{m} \left( \sqrt{2n} \|\mathbf{X}[j,:] - \mathbf{X}_w[j,:]\|_\infty \right)^k \right)^{1/k} \\
&\le \max_{j \in [m]} \left( \sqrt{2n} \|\mathbf{X}[j,:] - \mathbf{X}_w[j,:]\|_\infty \right) \\
&= \sqrt{2n} \|\mathbf{X} - \mathbf{X}_w\|_\infty \\
&\le \frac{\sqrt{n/2}}{b} \cdot \log_2\left( \frac{mn}{\delta} \right).
\end{aligned}
$$

$\square$

## B  Proofs of Detectability

**Proof of Lemma 5.1**

*Proof.* Let $I_j = \left[\frac{j-1}{b}, \frac{j}{b}\right]$ for all $j \in [b]$ denote an interval on $[0, 1]$. Define the indicator random variable $\mathbf{1}_g : [b] \to \{0, 1\}$ as – $\mathbf{1}_g(j) = 1$ if $I_j$ is labeled green. Then given $\mathbf{1}_g$, we know which intervals are green and which intervals are red. Also define $G = \cup_{j \in [b], \mathbf{1}_g(j)=1} I_j$ as the union of the intervals which are labeled green. So by definition $G$ is a set valued random variable. Therefore conditioned on $\mathbf{1}_g$ we have

$$\Pr[x \in G \mid \mathbf{1}_g] = \sum_{j \in [b], \mathbf{1}_g(j)=1} \Pr[x \in I_j] \tag{6}$$

$$= \sum_{j=1}^{b} \Pr[x \in I_j] \cdot \mathbf{1}_g(j) \tag{7}$$

where the second equality uses the fact that $\mathbf{1}_g(j) \in \{0, 1\}$ for all $j \in [b]$. Then, using the fact that the expected value of conditional probability of an event gives the probability of that event we have

$$\Pr(x \in G) = \mathbb{E}\left[\Pr[x \in G \mid \mathbf{1}_g]\right] \tag{8}$$

$$= \sum_{j=1}^{b} \Pr[x \in I_j] \cdot \mathbb{E}[\mathbf{1}_g(j)] \tag{9}$$

$$= \frac{1}{2} \sum_{j=1}^{b} \Pr[x \in I_j] \tag{10}$$

$$= \frac{1}{2}, \tag{11}$$

where in the second equality we use that the expected value of $\mathbf{1}_g(j)$ is the probability that $I_j$ is labeled green (this is equal to $1/2$ by assumption), and in the last equality we use the fact that $x$ is supported in $[0, 1]$.

$\square$

## C  Proofs of Robustness

**Proof of Theorem 6.2**

*Proof.* We first look at the number of preserved columns under uniform pairing.

**Uniform pairing.**  Let $\mathbf{X}'_w \in [0, 1]^{m \times k}$ denote the watermarked dataset after feature selection.

Consider the set of all possible pairs of columns after feature selection can be matched together. Then, there are $\binom{k}{2}$ possible choices, which are the number of 2-subset from $k$ total columns. Define an event $Y_\ell = \mathbf{1}[\ell\text{-th subset is a pair}]$. In each 2-subset, after a first column is selected, the probability that the second column actually forms a pair with the first column is $1/2n-1$. Hence, we have:

$$\mathbb{E}[Y_\ell] = \frac{1}{2n - 1}.$$

Hence, the expected number of preserved pairs of columns in $\mathbf{X}'_w$ under uniform random pairing can be obtained by summing over all $\binom{k}{2}$ possible choices as follow.

$$\mathbb{E}[\text{number of preserved pairs} \in \mathbf{X}'_w] = \sum_{\ell=1}^{\binom{k}{2}} \mathbb{E}[Y_\ell]$$

$$= \binom{k}{2} \frac{1}{2n - 1}$$

$$= \frac{(k-1)k}{2(2n-1)}$$

Now, we can look at the number of preserved pairs of columns under feature importance pairing. Note that we assume the feature selection is performed according to Assumption 6.1, and the columns are in descending order of feature importance. Since we are only retaining the top $k$ most important features after feature selection, this is equivalent to counting the number of preserved pairs among the first $k$ features after reordering.

**Feature importance pairing.** When the columns are paired according to the feature importance ordering, we match them according to Equation (4). Define the event $Y_{i,j} = \mathbf{1}[\text{columns } (i,j) \text{ is a pair}]$. Then, for fixed columns $i$ and $j$, we have:

$$\mathbb{E}[Y_\ell] = \frac{\frac{1}{|i-j|}}{\sum_{j\neq i}^{2n} \frac{1}{|i-\ell|}}$$

$$= \frac{\frac{1}{|i-j|}}{\sum_{\ell=1}^{i-1} \frac{1}{\ell} + \sum_{s=1}^{2n-i} \frac{1}{s}}$$

$$= \frac{\frac{1}{|i-j|}}{H_{i-1} + H_{2n-i}}$$

where $H_i$ is the $i$-th harmonic number. We know that for $i \geq 1$, we have $\ln(i+1) \leq H_i \leq \ln(i) + 1$. Let $\Delta_i = \max\{i-1, 2n-i\}$. Since only the top-k columns are retained, we have:

$$\mathbb{E}[\text{number of preserved pairs} \in \mathbf{X}'_w]$$

$$= \sum_{i=1}^{k} \sum_{j\neq i, j=1}^{k} \frac{1}{|i-j|\,(H_{i-1} + H_{2n-i})}$$

$$= \sum_{i=1}^{k} \frac{1}{H_{i-1} + H_{2n-i}} \left( \sum_{j\neq i, j=1}^{k} \frac{1}{|i-j|} \right)$$

$$= \sum_{i=1}^{k} \frac{1}{H_{i-1} + H_{2n-i}} (H_{i-1} + H_{k-i})$$

$$= \sum_{i=1}^{k} 1 - \frac{H_{2n-i} - H_{k-i}}{H_{i-1} + H_{2n-i}}$$

$$= k - \sum_{i=1}^{k} \frac{H_{2n-i} - H_{k-i}}{H_{i-1} + H_{2n-i}}$$

$$\geq k - \sum_{i=1}^{k} \frac{H_{2n-i}}{H_{i-1} + H_{2n-i}}$$

Since we assume that $k < n$, we have

$$i < n \Rightarrow 2n - i > n > i - 1$$

$$\Rightarrow \frac{1}{i-1} > \frac{1}{2n-i}$$

$$\Rightarrow H_{i-1} < H_{2n-i}$$

$$\Rightarrow \frac{H_{2n-i}}{H_{i-1} + H_{2n-i}} > \frac{1}{2}$$

Therefore, the expected number of preserved pairs under feature importance pairing is $k/2$.

Finally, we can compare the number of preserved pairs under the two pairing schemes:

$$
\frac{\frac{k}{2}}{\frac{(k-1)k}{2(2n-1)}} = \frac{k}{2}\frac{2(2n-1)}{(k-1)k}
$$
$$
= \frac{2n-1}{k-1}
$$
$$
\geq \frac{2k-1}{k-1}
$$
$$
> \frac{2k-2}{k-1}
$$
$$
= 2
$$

Therefore, we retain twice as many pairs of $(key, value)$ columns by using feature importance pairing scheme compared to the naive approach of uniformly random pairing. $\qquad\square$

**Proof of Theorem 6.3**

*Proof.* If $x_{\mathrm{tr}}$ falls out of the original green interval $I_j$, then we know that either $x_{\mathrm{tr}} > \frac{j}{b}$ or $x_{\mathrm{tr}} < \frac{j-1}{b}$. Since $b \leq 10^p$, we know that the green interval $I_j$ lies in the union of at most two consecutive grids. We consider two cases depending on whether $I_j$ contains a grid point or not.

**When $I_j$ does not contain a grid point.** Since the truncation function defined in Equation (5) always truncate an element $x$ to the nearest left grid point, which lies outside of $I_j$, we have

$$
\Pr\left[x_{\mathrm{tr}} \notin I_j | c \notin I_j, \forall c \in \mathrm{grid}\right] = 1.
$$

**When $I_j$ contains a grid point.** Let $c$ denote the grid point in $I_j$. Then, when the interval $I_j$ contains a grid point, we have:

$$
\Pr[x_{\mathrm{tr}} \notin I_j | c \in I_j] = \Pr\left[x \in \left[\frac{j-1}{b}, c\right)\right]
$$
$$
= \frac{c - \frac{j-1}{b}}{\frac{1}{b}}
$$
$$
= c \cdot b - j + 1
$$

Then, summing over all possible events, we have the probability that the truncation attack successfully moves a watermarked element out of its original green interval is:

$$
\Pr[x_{\mathrm{tr}} \notin I_j]
$$
$$
= \Pr[x_{\mathrm{tr}} \notin I_j | c \notin I_j, \forall c \in \mathrm{grid}] \cdot \Pr[c \notin I_j, \forall i \in \mathrm{grid}]
$$
$$
+ \Pr[x_{\mathrm{tr}} \notin I_j | c \in I_j] \cdot \Pr[c \in I_j]
$$
$$
= \Pr[c \notin I_j, \forall c \in \mathrm{grid}] + (c \cdot b - j + 1)\Pr[c \in I_j]
$$
$$
= \left(\frac{b-1}{b}\right)^{10^p} + \frac{c \cdot b - j + 1}{b}
$$
$$
= \frac{(b-1)^{10^p} + b^{10^p - 1}(c \cdot b - j + 1)}{b^{10^p}}
$$

Hence, the probability that $x_{\mathrm{tr}}$ is in a red interval is $\rho_{\mathrm{trunc}} = \frac{(b-1)^{10^p} + b^{10^p-1}(c \cdot b - j + 1)}{2b^{10^p}}$.

Let $n_\alpha = \alpha\sqrt{\frac{m}{4}} + \frac{m}{2}$ be the minimum nmber of cells in the green intervals for the $z$-score to be at least $\alpha$. For the watermark to be removed, the attacker needs to truncate at least $m - n_\alpha$ cells in a column. Therefore, the expected number of attacked cells is $\frac{m - n_\alpha}{\rho_{\mathrm{trunc}}}$. $\qquad\square$

### C.1 Robustness to additive noise

**Theorem C.1** (Robustness under noise). *Fix a column in the watermarked dataset. Let $n_a$ be the number of cells in this column that an adversary can inject noise into. Let $S$ be a subset of $[b]$ such that $|S| = n_a$. Also let the noise injected into each cell is drawn i.i.d. from some known distribution, i.e., $\epsilon \sim \mathcal{D}$.*

*Define parameter $\gamma$ as*

$$\gamma := \frac{1}{n_a} \sum_{i \in S} \sum_{j \in [b]} \Pr[(x_i + \epsilon_i)^\circ \in I_j], \tag{12}$$

*where $\gamma \leq 1$ and $\epsilon_i$ is the noise added to $x_i$. Then, we must have*

$$n_a \geq \frac{m - \alpha\sqrt{m}}{2 - \gamma}, \tag{13}$$

*for the expected $z$-score to be less than $\alpha$ (i.e., to remove the watermark).*

*Proof.* Define $x^\circ = x - \lfloor x \rfloor$ as the fractional part of any $x \in \mathbb{R}$.

Let $I_j = \left[\frac{j-1}{b}, \frac{j}{b}\right]$ for all $j \in [b]$ denote an interval on $[0, 1]$. Define the indicator random variable $\mathbf{1}_g : [b] \to \{0, 1\}$ as $\mathbf{1}_g(j) = 1$ if $I_j$ is labeled green. Then given $\mathbf{1}_g$, we know which intervals are green and which intervals are red. Also define $G = \cup_{j \in [b], \mathbf{1}_g(j)=1} I_j$ as the union of the intervals which are labeled green. So by definition $G$ is a set valued random variable.

Furthermore, we define $S$ to be a subset of row indices, $[m]$, in which we inject noise, such that $|S| = n_a$. For each $i \in S$, let $\epsilon_i$ be i.i.d. samples drawn from $\mathcal{D}$, and for each $i \notin S$, $\epsilon_i = 0$.

We want to find a bound on $n_a$ such that in expectation the number of cells in a column that are watermarked is below the required $z$-score threshold. Given $z$-score threshold equal to $\alpha$, the minimum number of cells that need to be in the green intervals is $T_0 = \frac{\alpha\sqrt{m}+m}{2}$. The expected number of cells in the column where we add noise that is still in a green interval is

$$\mathbb{E}\left[\sum_{i \in S} \mathbf{1}[(x_i + \epsilon_i)^\circ \in G] \mid \mathbf{1}_g\right] = \sum_{i \in S} \Pr\left[(x_i + \epsilon_i)^\circ \in G \mid \mathbf{1}_g\right]. \tag{14}$$

For all $i \in S$, conditioned on $\mathbf{1}_g$ we have

$$\Pr\left[(x_i + \epsilon_i)^\circ \in G \mid \mathbf{1}_g\right] = \sum_{j \in [b], \mathbf{1}_g(j)=1} \Pr[(x_i + \epsilon_i)^\circ \in I_j] \tag{15}$$

$$= \sum_{j=1}^{b} \Pr[(x_i + \epsilon_i)^\circ \in I_j] \cdot \mathbf{1}_g(j), \tag{16}$$

where the second equality uses the fact that $\mathbf{1}_g(j) \in \{0, 1\}$ for all $j \in [b]$.

From Equation (14) and Equation (16) and taking expectation over the labeling of the intervals we have

$$\mathbb{E}\left[\sum_{i \in S} \mathbf{1}[(x_i + \epsilon_i)^\circ \in G]\right] = \mathbb{E}\left[\mathbb{E}\left[\sum_{i \in S} \mathbf{1}[(x_i + \epsilon_i)^\circ \in G] \mid \mathbf{1}_g\right]\right] \tag{17}$$

$$= \sum_{i \in S} \sum_{j \in [b]} \Pr[(x_i + \epsilon_i)^\circ \in I_j] \cdot \mathbb{E}\left[\mathbf{1}_g(j)\right] \tag{18}$$

$$= \frac{n_a \gamma}{2}, \tag{19}$$

where the last line uses the definition of $\gamma$ in Equation (12).

Now we prove that $0 \leq \gamma \leq 1$. Since $\gamma$ is the sum of probabilities, it is non-negative. Since,

$$\gamma = \frac{1}{n_a} \sum_{i \in S} \sum_{j \in [b]} \Pr[(x_i + \epsilon_i)^\circ \in I_j], \tag{20}$$

and for every $i \in S$, there is exactly one $j \in [b]$ such that $(x_i + \epsilon_i)^\circ \in I_j$. So, we must have $\gamma \leq 1$.

The expected number of cells which are in a green interval within the column is then:

$$m - n_a + \frac{n_a \gamma}{2}. \tag{21}$$

So, to break the watermark Equation (21) should be less than $T_0$, which gives

$$m - n_a + \frac{n_a \gamma}{2} \leq \frac{\alpha\sqrt{m} + m}{2} \tag{22}$$

$$n_a \geq \frac{m - \alpha\sqrt{m}}{2 - \gamma}. \tag{23}$$

$\square$

**Remark C.2** (Distribution independent robustness). *We observe that since $0 \leq \gamma \leq 1$, the lower bound on $n_a$ can also be expressed independent of $\gamma$ as*

$$n_a \geq \frac{1}{2}\left(m - \alpha\sqrt{m}\right). \tag{24}$$

*Thus, even when the distribution of the noise is not known, one can have an estimate on the maximum number of cells in a column of the watermarked table that needs to be corrupted to remove the watermark.*

**Corollary C.3** (Robustness under uniform noise). *Fix a column in the watermarked dataset. Let $n_a$ be the number of cells in this column that an adversary can inject noise into. Let $S$ be a subset of $[b]$ such that $|S| = n_a$. Also let the noise injected into each cell is drawn i.i.d from an uniform distribution, i.e., $\epsilon \sim \mathsf{Unif}[-\sigma, \sigma]$ for $0 < \sigma \leq 1$.*

*Define parameter $\gamma$ as*

$$\gamma := \frac{1}{2n_a\sigma} \sum_{i \in S} \sum_{j=1}^{b} \left( \max\left(0, \min\left(\frac{j}{b}, x_i + \sigma\right) - \max\left(\frac{j-1}{b}, x_i - \sigma\right)\right) \right.$$

$$+ \max\left(0, \min\left(\frac{j}{b} - 1, x_i + \sigma\right) - \max\left(\frac{j-1}{b} - 1, x_i - \sigma\right)\right)$$

$$\left. + \max\left(0, 1 + \min\left(\frac{j}{b}, x_i + \sigma\right) - \max\left(1 + \frac{j-1}{b}, x_i - \sigma\right)\right) \right), \tag{25}$$

*where $\gamma \leq 1$. Then, we must have*

$$n_a \geq \frac{m - \alpha\sqrt{m}}{2 - \gamma}, \tag{26}$$

*for the expected z-score to be less than $\alpha$ (i.e., to remove the watermark).*

*Proof.* We just need to show Equation (25) holds, since the remaining claims of the statement follow directly from Theorem C.1. There are the following three possible cases for $(x_i + \epsilon_i)^\circ$ using $\epsilon_i \sim \mathsf{Unif}[-\sigma, \sigma]$

$$(x_i + \epsilon_i)^\circ = \begin{cases} 1 + x_i + \epsilon_i & x_i + \epsilon_i \in [-1, 0) \\ x_i + \epsilon_i & x_i + \epsilon_i \in [0, 1] \\ x_i + \epsilon_i - 1 & x_i + \epsilon_i \in (1, 2]. \end{cases} \tag{27}$$

Then using the union bound we have,

$$\Pr[(x_i + \epsilon_i)^\circ \in I_j] = \Pr\left[\frac{j-1}{b} - 1 \le x_i + \epsilon_i \le \frac{j}{b} - 1\right] + \Pr\left[\frac{j-1}{b} \le x_i + \epsilon_i \le \frac{j}{b}\right] +$$
$$\Pr\left[1 + \frac{j-1}{b} \le x_i + \epsilon_i \le 1 + \frac{j}{b}\right]. \tag{28}$$

**Equating Equation (28).** Since $e_i \sim \mathsf{Unif}[-\sigma, \sigma]$, then $x_i + e_i \sim \mathsf{Unif}[x_i - \sigma, x_i + \sigma]$. Then we have

$$\Pr\left[\frac{j-1}{b} \le x_i + \epsilon_i \le \frac{j}{b}\right] = \frac{\max\left(0, \min\left(\frac{j}{b}, x_i + \sigma\right) - \max\left(\frac{j-1}{b}, x_i - \sigma\right)\right)}{2\sigma}, \tag{29}$$

where $\max\left(0, \min\left(\frac{j}{b}, x_i + \sigma\right) - \max\left(\frac{j-1}{b}, x_i - \sigma\right)\right)$ is the overlap between the intervals $[x_i - \sigma, x_i + \sigma]$ and $\left[\frac{j-1}{b}, \frac{j}{b}\right]$.

Similarly, we have

$$\Pr\left[\frac{j-1}{b} - 1 \le x_i + \epsilon_i \le \frac{j}{b} - 1\right] = \frac{\max\left(0, \min\left(\frac{j}{b} - 1, x_i + \sigma\right) - \max\left(\frac{j-1}{b} - 1, x_i - \sigma\right)\right)}{2\sigma} \tag{30}$$

$$\Pr\left[1 + \frac{j-1}{b} \le x_i + \epsilon_i \le 1 + \frac{j}{b}\right] = \frac{\max\left(0, 1 + \min\left(\frac{j}{b}, x_i + \sigma\right) - \max\left(1 + \frac{j-1}{b}, x_i - \sigma\right)\right)}{2\sigma} \tag{31}$$

**Combining everything.** Plugging Equation (30), Equation (31), and Equation (29) in Equation (28) and subsequently in Equation (12) gives us the claim. As stated before, the rest of the proof follows from Theorem C.1. $\qquad\square$

**Corollary C.4** (Robustness under Gaussian noise)**.** *Fix a column in the watermarked dataset. Let $n_a$ be the number of cells in this column that an adversary can inject noise into. Let $S$ be a subset of $[b]$ such that $|S| = n_a$. Also let the noise injected into each cell is drawn i.i.d from an Gaussian distribution, i.e., $\epsilon \sim \mathcal{N}(0, \sigma^2)$ for $0 < \sigma \le 1$.*

*Define parameter $\gamma$ as*

$$\gamma := \frac{1}{n_a} \sum_{i \in S} \sum_{j=1}^{b} \frac{1}{\sqrt{2\pi}\sigma} \int_{\frac{j-1}{b}}^{\frac{j}{b}} \sum_{k=-\infty}^{\infty} \exp\left(-\frac{(\theta/2\pi - x_i + 2\pi k)^2}{2\sigma^2}\right) d\theta, \tag{32}$$

*where $\gamma \le 1$ and $\epsilon_i$ is the noise added to $x_i$. Then, we must have*

$$n_a \ge \frac{m - \alpha\sqrt{m}}{2 - \gamma}, \tag{33}$$

*for the expected z-score to be less than $\alpha$ (i.e., to remove the watermark).*

*Proof.* Similar to Corollary C.3 we only need to show Equation (32) holds, since the rest of the proof follows from Theorem C.1. Observe that when $\epsilon_i \sim \mathcal{N}(0, \sigma^2)$, $(x_i + \epsilon_i)^\circ$ can be expressed as:

$$(x_i + \epsilon_i)^\circ = x_i + \epsilon_i - \lfloor x_i + \epsilon_i \rfloor. \tag{34}$$

Observe that the random variable $(x_i + \epsilon_i)^\circ$ is always the fractional part of $x_i + \epsilon_i$, and $x_i + \epsilon_i \sim \mathcal{N}(x_i, \sigma^2)$. The distribution of $(x_i + \epsilon_i)^\circ$ is exactly the wrapped normal distribution (Jammalamadaka & Sengupta, 2001, §2.2.6) with the density function

$$f(\theta; x_i, \sigma) = \frac{1}{\sqrt{2\pi}\sigma} \sum_{k=-\infty}^{\infty} \exp\left(-\frac{(\theta/2\pi - x_i + 2\pi k)^2}{2\sigma^2}\right), \tag{35}$$

and is obtained by wrapping the normal distribution around the unit circle, and rescaling the support of the random variable to $[0, 1]$. Then we have

$$\Pr\left[\frac{j-1}{b} \leq (x_i + \epsilon_i)^\circ \leq \frac{j}{b}\right] = \frac{1}{\sqrt{2\pi}\sigma} \int_{\frac{j-1}{b}}^{\frac{j}{b}} \sum_{k=-\infty}^{\infty} \exp\left(-\frac{(\theta/2\pi - x_i + 2\pi k)^2}{2\sigma^2}\right) d\theta. \tag{36}$$

Since $\Pr\left[\frac{j-1}{b} \leq (x_i + \epsilon_i)^\circ \leq \frac{j}{b}\right] = \Pr\left[(x_i + \epsilon_i)^\circ \in I_j\right]$, using the definition of $\gamma$ we have our claim. $\qquad\square$

**Remark C.5.** *While Equation* (32) *looks ominous, it has been extensively studied in the literature. Importantly there are analytical expressions (that avoids the infinite sum, e.g., von Mises distribution closely matches the density function of wrapped normal distribution (Mardia & Jupp, 2009)) that can approximate Equation* (35) *well. Furthermore, one can also empirically approximate Equation* (32) *by first approximating the infinite sum using small values of k (Kurz et al., 2014) and then evaluating the integral.*

# D    Additional experimental details

For synthetic data generation using three generative methods mentioned in the paper, we follow the default hyperparameters found in the SDV library (Patki et al., 2016). We run the experiments on a machine type of g4dn.4xlarge consisting of 16 CPU, 64GB RAM, and 1 GPU. For Utility evaluation, default parameters of the XGBoost classifier (Chen & Guestrin, 2016) and Random Forest classifier with a seed of 42 was used. The synthetic data generation, training and evaluation process typically finishes within 4 hours. Python 3.8 version was used to run the experiments.

## D.1    Standard deviation

To investigate the performance of our tabular watermark on real-world data, we sample from each generative model a generated dataset. For each setup, we create 5 watermarked training sets of each generator so as to measure the mean accuracy and standard deviation. We use XGBoost classifier and Random Forest classifier for classification tasks which are then evaluated on real testing sets.

| Dataset | Method | Not WM | Watermarked (WM) | | | |
|---------|--------|--------|-----|-----|-----|-----|
| | | | WM | WM and Truncated | WM and 40% cols drop | |
| | | | | | FI | Random |
| California | CTGAN | $0.373 \pm 0.02$ | $0.371 \pm 0.01$ | $0.368 \pm 0.01$ | $0.203 \pm 0.1$ | $0.256 \pm 0.05$ |
| | Copula | $0.371 \pm 0.02$ | $0.376 \pm 0.03$ | $0.376 \pm 0.03$ | $0.31 \pm 0.1$ | $0.30 \pm 0.13$ |
| | TVAE | $0.797 \pm 0.02$ | $0.799 \pm 0.02$ | $0.798 \pm 0.02$ | $0.385 \pm 0.19$ | $0.365 \pm 0.08$ |
| Wilt | CTGAN | $0.731 \pm 0.03$ | $0.733 \pm 0.04$ | $0.733 \pm 0.04$ | $0.563 \pm 0.03$ | $0.563 \pm 0.03$ |
| | Copula | $0.99 \pm 0.01$ | $0.996 \pm 0.01$ | $0.996 \pm 0.01$ | $0.993 \pm 0.02$ | $0.993 \pm 0.02$ |
| | TVAE | $0.989 \pm 0.00$ | $0.989 \pm 0.00$ | $0.989 \pm 0.00$ | $0.972 \pm 0.04$ | $0.803 \pm 0.01$ |

Table 3: Accuracy of the downstream models under various attacks to the watermarked datasets. In particular, we provide accuracy for the original dataset and its watermarked counterpart. Classifier used for utility evaluation is **XGBoost.** In addition to this, we add the standard deviation of each record.

## D.2    Detection computation

We further investigate the effects that algorithm parameters and downstream manipulations have on the robustness and computational requirements of the detection mechanism. We measure the number of column pairing tests that must be done before a high confidence pair is found. We define high confidence as achieving a $z$-score of 4 using 24 randomly selected rows. The process is stopped early when the watermark is detected. Thus, when a watermark cannot be found, the result becomes $N^2 - N$, where $N$ is the number of columns in the dataset.

| Dataset | Method | Not WM | Watermarked (WM) | | | |
| | | | WM | WM and Truncated | WM and 40% cols drop | |
| | | | | | FI | Random |
|---|---|---|---|---|---|---|
| California | CTGAN | $0.379 \pm 0.01$ | $0.379 \pm 0.01$ | $0.382 \pm 0.01$ | $0.309 \pm 0.13$ | $0.334 \pm 0.12$ |
| | Copula | $0.377 \pm 0.02$ | $0.393 \pm 0.03$ | $0.393 \pm 0.03$ | $0.278 \pm 0.18$ | $0.333 \pm 0.07$ |
| | TVAE | $0.805 \pm 0.02$ | $0.809 \pm 0.02$ | $0.808 \pm 0.02$ | $0.399 \pm 0.19$ | $0.449 \pm 0.18$ |
| Wilt | CTGAN | $0.849 \pm 0.04$ | $0.856 \pm 0.04$ | $0.859 \pm 0.04$ | $0.644 \pm 0.40$ | $0.617 \pm 0.42$ |
| | Copula | $0.986 \pm 0.01$ | $0.986 \pm 0.01$ | $0.985 \pm 0.01$ | $0.973 \pm 0.02$ | $0.969 \pm 0.02$ |
| | TVAE | $0.985 \pm 0.00$ | $0.985 \pm 0.00$ | $0.985 \pm 0.00$ | $0.985 \pm 0.00$ | $0.81 \pm 0.40$ |

Table 4: Accuracy of the downstream models under various attacks to the watermarked datasets. In particular, we provide accuracy for the original dataset and its watermarked counterpart. Classifier used for utility evaluation is **Random Forest.** In addition to this, we add the standard deviation of each record.

| Dataset | Method | Not watermarked | Watermarked | Watermarked & truncated |
|---|---|---|---|---|
| California | CTGAN | $72 \pm 0$ | $16 \pm 0$ | $19.6 \pm 4.41$ |
| | Copula | $72 \pm 0$ | $16 \pm 0$ | $30 \pm 4.94$ |
| | TVAE | $72 \pm 0$ | $58.6 \pm 11.11$ | $58.4 \pm 11.35$ |
| Wilt | CTGAN | $20 \pm 0$ | $9.2 \pm 4.62$ | $9.2 \pm 4.62$ |
| | Copula | $20 \pm 0$ | $5.4 \pm 3.5$ | $5.6 \pm 3.38$ |
| | TVAE | $20 \pm 0$ | $2.4 \pm 0.5$ | $2.2 \pm 0.4$ |

Table 5: Number of column pair tests executed during detection process. The process is stopped early when the watermark is detected. Thus, when a watermark cannot be found, the result becomes $N^2 - N$, where $N$ is the number of columns in the dataset.

We compare non-watermarked and watermarked data, and also test with truncation and dropped columns as before. For the latter, here, we compare the 4 total combinations of column ordering choices: watermarking and detection both present the option to order by feature importance or to order randomly. We find that, with truncation, it generally takes the same number of computations as without, and that this amount is significantly lower than the upper bound in the 'Not watermarked' column.

### D.3 Comparison to prior work

We investigate the performance of our watermarking technique on four different real-world datasets and compare it with prior watermarking algorithms as benchmark.

**Experimental Setup.** We evaluate our watermark and the baseline approaches on four real-world datasets: **Boston Housing** and **California Housing Price** for regression task, as well as **Adult** and **Wilt** for classification task. For each dataset, we generate corresponding synthetic datasets using Gaussian Copula (Masarotto & Varin, 2012). We use Random Forest as the downstream machine learning algorithm for all datasets.

**Baseline approaches.** We compare our performance to three other tabular watermarking techniques: WGTD (He et al., 2024), TabularMark (Zheng et al., 2024), and RINTAW (Fang et al., 2025). As the source codes are unavailable, we implement these watermarking techniques using the authors' specifications. That is, for WGTD implementation, we set the number of 'green list' intervals to be $m = 500$, for a total of 1000 intervals between 0 and 1. For TabularMark implementation, we always embed the watermark into 10% of the label column, with number of unit domains $k = 500$, perturbation range $p = 25$. For classification task, we select a half of the possible categories to be 'green' instead of using the aforementioned unit domains and perturbation range. Finally, for RINTAW implementation, we use the default masking ratio of 0.

**Experimental results.** First, we study the fidelity of our watermarking approach on these datasets. Particularly, we are interested in measuring both downstream performance metric (RMSE for regression task and ROC-AUC score for classification task) and the Machine Learning Efficiency (MLE), i.e., the gap between the performance of watermarked data and the real test data. Each experiment is repeated 10 times and the mean and standard deviation for the target performance metric are reported. The result of this experiment is summarized in Table 6. Overall, RINTAW achieves the best downstream performance among all watermarking methods. However, our approach achieves the smallest MLE Gap in three datasets (Boston Housing, Adult, and Wilt), and the second smallest MLE gap in the remaining dataset (California Housing). That is, for both regression and classification, our watermark approach leads to minimal distortion compared to the real, unwatermarked dataset.

| | Boston | | California | | Adult | | Wilt | |
|---|---|---|---|---|---|---|---|---|
| | RMSE | MLE Gap | RMSE | MLE Gap | ROC-AUC | MLE Gap | ROC-AUC | MLE Gap |
| **Ours** | $0.330 \pm 0.000$ | **0.000** | $84586.003 \pm 74.140$ | 74.459 | $0.741 \pm 0.000$ | **0.000** | $0.496 \pm 0.000$ | **0.000** |
| WGTD | $0.334 \pm 0.005$ | 0.004 | $84832.096 \pm 220.436$ | 319.557 | $0.742 \pm 0.004$ | 0.004 | $0.496 \pm 0.000$ | **0.000** |
| TabularMark | $0.987 \pm 0.048$ | 0.656 | $\mathbf{84521.708 \pm 59.071}$ | **9.164** | $0.742 \pm 0.004$ | 0.001 | $0.534 \pm 0.157$ | 0.037 |
| RINTAW | $\mathbf{0.328 \pm 0.002}$ | 0.001 | $85047.119 \pm 601.940$ | 534.575 | $\mathbf{0.738 \pm 0.004}$ | 0.003 | $\mathbf{0.493 \pm 0.002}$ | 0.003 |

Table 6: We report the RMSE for regression task and ROC-AUC score for classification task. Each experiment is repeated 10 times and the mean and standard deviation are reported. Furthermore, we also report the MLE gap, which is the difference between the downstream performance of watermarked data and real test data. The best results in each column is highlighted in bold.

Second, we investigate the detectability and robustness of our watermark under different threat models. Particularly, we employ eight common operations (attacks) on tabular data, divided into two categories: (1) table level attacks: row shuffling, row deletion, column deletion, feature selection according to feature importance; and (2) cell level attacks: Gaussian noise addition, uniform noise addition, value alteration, and truncation. For row deletion, we randomly remove 20% of rows. For column deletion, we randomly remove 2 feature columns. For feature selection according to feature importance, we keep the top 40% of columns with the highest feature importance (detailed in Section 6.1). For Gaussian noise addition, we inject i.i.d. Gaussian noise with mean zero and variance one into numeric columns. For uniform noise addition, we inject noise drawn i.i.d. from $\mathsf{Unif}[-0.1, 0.1]$ into numeric columns. For value alteration, we modify the numeric columns by multiplying them with a random factor drawn uniformly at random from $[0.9, 1.1]$, i.e., increase or decrease each numeric cell by at most 10% of their original value. For truncation, we truncate all numeric float values to only 2 decimal places. For each attack, we report the mean $p$-value from the detection procedure, where the dataset is detected as watermarked if $p$-value $< 0.05$. The result of this experiment is summarized in Table 7. Among all considered watermarking techniques, our method is the most consistent at detecting the embedded watermark under all threat models. On the other hand, WTGD is generally robust to table level attacks (row deletion, column deletion, feature importance column deletion), but is more susceptible to additive noise attacks. TabularMark is the least robust method, as it often fails to detect watermark under most cell level attacks (additive noise, value alteration, and truncation). Finally, RINTAW is very robust to row deletion and row shuffling, but may fail to detect the watermark under column deletion and cell level attacks. Overall, our watermarking technique achieves both minimal distortion (small MLE gap in Table 6) and high fidelity/robustness (consistently low $p$-value in Table 7).

# E    Decoding algorithms

Consider the case that the continuous space in $(0, 1]$ is uniformly discretized using bins of size $1/b$ i.e., construct bins in $(0, 1]$ as $-\left\{(0, \frac{1}{b}], (\frac{1}{b}, \frac{2}{b}], \ldots, (\frac{b-1}{b}, 1]\right\}$. Then each of these bins are labeled $\{0, 1\}$ with some probability $1 - p_i$ and $p_i$ respectively. We save this data-structure. During query time, each entry of the input table/row is checked to see if the values fall into the bins labeled 1. One can then compute $z$-scores based on the count of watermarked data identified in the input table. The output of the query function is $\{0, 1\}$ if the table is not watermarked or watermarked, respectively.

| Dataset | Method | Watermarked | Row Shuffled | Row Del. | Col. Del. | Importance | Gaussian | Uniform | Alteration | Truncation |
|---|---|---|---|---|---|---|---|---|---|---|
| Boston | **Ours** | $0.007 \pm 0.000$ | $0.012 \pm 0.003$ | $0.011 \pm 0.008$ | $0.011 \pm 0.007$ | $0.007 \pm 0.000$ | $0.018 \pm 0.008$ | $0.018 \pm 0.008$ | $0.018 \pm 0.009$ | $0.002 \pm 0.000$ |
| | WTGD | $0.000 \pm 0.000$ | $0.000 \pm 0.000$ | $0.000 \pm 0.000$ | $0.000 \pm 0.000$ | $1.000 \pm 0.490$ | $0.462 \pm 0.011$ | $0.418 \pm 0.015$ | $0.000 \pm 0.000$ | $0.000 \pm 0.000$ |
| | TabularMark | $0.000 \pm 0.000$ | $0.000 \pm 0.000$ | $0.000 \pm 0.000$ | $1.000 \pm 0.400$ | $1.000 \pm 0.000$ | $1.000 \pm 0.000$ | $1.000 \pm 0.000$ | $1.000 \pm 0.000$ | $1.000 \pm 0.000$ |
| | RINTAW | $0.000 \pm 0.000$ | $0.000 \pm 0.000$ | $0.000 \pm 0.000$ | $0.000 \pm 0.000$ | $0.529 \pm 0.212$ | $0.594 \pm 0.137$ | $0.581 \pm 0.381$ | $0.386 \pm 0.315$ | $0.000 \pm 0.000$ |
| California | **Ours** | $0.000 \pm 0.000$ | $0.000 \pm 0.000$ | $0.000 \pm 0.000$ | $0.000 \pm 0.000$ | $0.000 \pm 0.000$ | $0.014 \pm 0.008$ | $0.018 \pm 0.005$ | $0.013 \pm 0.007$ | $0.000 \pm 0.000$ |
| | WTGD | $0.000 \pm 0.000$ | $0.000 \pm 0.000$ | $0.000 \pm 0.000$ | $0.000 \pm 0.000$ | $0.000 \pm 0.000$ | $0.482 \pm 0.006$ | $0.273 \pm 0.161$ | $0.055 \pm 0.100$ | $0.000 \pm 0.000$ |
| | TabularMark | $0.000 \pm 0.000$ | $0.000 \pm 0.000$ | $0.000 \pm 0.000$ | $1.000 \pm 0.000$ | $1.000 \pm 0.000$ | $1.000 \pm 0.000$ | $1.000 \pm 0.000$ | $1.000 \pm 0.000$ | $0.000 \pm 0.000$ |
| | RINTAW | $0.000 \pm 0.000$ | $0.000 \pm 0.000$ | $0.000 \pm 0.000$ | $0.000 \pm 0.000$ | $0.022 \pm 0.027$ | $0.520 \pm 0.325$ | $0.820 \pm 0.105$ | $0.442 \pm 0.046$ | $0.000 \pm 0.000$ |
| Adult | **Ours** | $0.000 \pm 0.000$ | $0.000 \pm 0.000$ | $0.000 \pm 0.000$ | $0.000 \pm 0.000$ | $0.000 \pm 0.000$ | $0.014 \pm 0.008$ | $0.010 \pm 0.009$ | $0.018 \pm 0.005$ | $0.000 \pm 0.000$ |
| | WTGD | $0.000 \pm 0.000$ | $0.000 \pm 0.000$ | $0.000 \pm 0.000$ | $0.000 \pm 0.000$ | $0.000 \pm 0.000$ | $0.446 \pm 0.012$ | $0.376 \pm 0.186$ | $0.000 \pm 0.000$ | $0.000 \pm 0.000$ |
| | TabularMark | $0.039 \pm 0.039$ | $0.358 \pm 0.277$ | $1.000 \pm 0.000$ | $1.000 \pm 0.000$ | $1.000 \pm 0.000$ | $1.000 \pm 0.000$ | $1.000 \pm 0.000$ | $1.000 \pm 0.000$ | $0.398 \pm 0.390$ |
| | RINTAW | $0.000 \pm 0.000$ | $0.000 \pm 0.000$ | $0.000 \pm 0.000$ | $0.000 \pm 0.000$ | $0.358 \pm 0.150$ | $0.561 \pm 0.313$ | $0.530 \pm 0.246$ | $0.667 \pm 0.149$ | $0.000 \pm 0.000$ |
| Wilt | **Ours** | $0.000 \pm 0.000$ | $0.004 \pm 0.008$ | $0.010 \pm 0.009$ | $0.087 \pm 0.029$ | $0.000 \pm 0.000$ | $0.014 \pm 0.008$ | $0.017 \pm 0.007$ | $0.038 \pm 0.040$ | $0.021 \pm 0.000s$ |
| | WTGD | $0.034 \pm 0.016$ | $0.338 \pm 0.167$ | $0.345 \pm 0.168$ | $0.381 \pm 0.147$ | $1.000 \pm 0.000$ | $0.483 \pm 0.014$ | $0.481 \pm 0.010$ | $0.486 \pm 0.006$ | $0.090 \pm 0.124$ |
| | TabularMark | $0.000 \pm 0.000$ | $0.000 \pm 0.000$ | $0.000 \pm 0.000$ | $1.000 \pm 0.000$ | $1.000 \pm 0.000$ | $1.000 \pm 0.000$ | $1.000 \pm 0.000$ | $1.000 \pm 0.000$ | $1.000 \pm 0.000$ |
| | RINTAW | $0.000 \pm 0.000$ | $0.000 \pm 0.000$ | $0.000 \pm 0.000$ | $0.000 \pm 0.000$ | $0.626 \pm 0.130$ | $0.584 \pm 0.340$ | $0.363 \pm 0.342$ | $0.440 \pm 0.280$ | $0.700 \pm 0.109$ |

Table 7: We report the $p$-value of each detection algorithm for different threat models and datasets. Each experiment is repeated 10 times and the mean and standard deviation are reported. Except for Wilt dataset and column deletion attack, our watermark can always be reliably detected for the remaining datasets and threat models, i.e., $p < 0.05$. On the other hand, WTGD often fails to detect the watermark under Gaussian noise addition and uniform noise addition. Furthermore, TabularMark fails to detect the watermark in most cell level attacks (additive noise, value alteration, and truncation). Finally, RINTAW may fail to detect the watermark under feature importance column drop and cell level attacks.

For a given input to the query function and the output watermark label, we formalize this problem as a learning problem as follows.

**Assumptions.** We assume that the number of bins $b$ is known to the detection algorithm. We make such an assumption because we use a statistical learning algorithm for detection, and if $b$ is not known apriori, the VC dimension of the relevant hypothesis class becomes infinity, giving us vacuous sample complexity bounds. The algorithm generating the watermark fixes a probability $p \in (0, 1)$, which is hidden from the detection algorithm. The watermarking algorithm draws $b$ independent and identically distributed samples from the Bernoulli distribution with parameter $p$. Let $y_1, \ldots, y_b \in \{0, 1\}$ denote the observed values. For $k \in [b]$, we interpret $y_k$ to be the label of the interval (or bin) $\left( \frac{k-1}{b}, \frac{k}{b} \right]$.

**Query function.** Let $\Pi \colon (0, 1] \to [b]$ denote the canonical projection onto the bins, defined as $\Pi(x) = k$ if $x \in \left( \frac{k-1}{b}, \frac{k}{b} \right]$ for $k \in [b]$. Clearly, $\Pi$ determines the index of the bin into which the input falls. Given a tabular data $\mathbf{X} \in (0, 1]^{m \times n}$ as input, we denote $\Pi^{m \times n} \colon (0, 1]^{m \times n} \to [b]^{m \times n}$ to be the function that implements $\Pi$ entry wise on $\mathbf{X}$. Let $F_z \colon [b]^{m \times n} \to \mathbb{R}$ be a $z$-score function (chosen based on the problem), which first maps the index of the bin in each entry of the tabular data to the corresponding label of that bin, and then processes these labels in an appropriate fashion.

We define the *query function* $Q \colon (0, 1]^{m \times n} \to \{0, 1\}$ given by $Q(\boldsymbol{x}) = \mathrm{sgn}(F_z(\Pi^{m \times n}(\boldsymbol{x})))$, where $\mathrm{sgn} \colon \mathbb{R} \to \{0, 1\}$ is the sign (or indicator) function defined as

$$\mathrm{sgn}(x) = \begin{cases} 1 & \text{if } x > 0 \\ 0 & \text{otherwise.} \end{cases} \tag{37}$$

The query function tells us whether (1) or not (0) the input data is watermarked. Since the labels $y_k$ are not known to the detection algorithm, the query function is also not known.

**Goal.** We want to approximate the query function up to a small prediction error with high probability. Suppose that $\mathbf{X}$ is the data random variable, taking values in $(0, 1]^{m \times n}$. We are given $M$ independent and identically distributed training samples $(\mathbf{X}_1, Q(\mathbf{X}_1)), \ldots, (\mathbf{X}_M, Q(\mathbf{X}_M))$. We wish to use these training samples to learn the function $Q$, such that given $\epsilon \in (0, 1)$ and $\delta \in (0, 1)$, we have

$$\Pr_{\mathbf{X}_1, \ldots, \mathbf{X}_M} \left( \Pr_{\mathbf{X}'} \left( \mathcal{A}((\mathbf{X}_i)_{i=1}^M, \mathbf{X}') \neq Q(\mathbf{X}') \right) \leq \epsilon \right) \geq 1 - \delta, \tag{38}$$

where $\mathcal{A}\colon ((0,1]^{m\times n})^M \times (0,1]^{m\times n} \to \{0,1\}$ is an algorithm that takes the training data $\mathbf{X}_1,\ldots,\mathbf{X}_M \in (0,1]^{m\times n}$ as input and outputs the query function $\mathcal{A}((\mathbf{X}_i)_{i=1}^M, \cdot)$. Given training data $\mathbf{X}_1,\ldots,\mathbf{X}_M \in (0,1]^{m\times n}$, the quantity $\Pr_{\mathbf{X}'}\left(\mathcal{A}((\mathbf{X}_i)_{i=1}^M, \mathbf{X}') \neq Q(\mathbf{X}')\right)$ is the expected value of the 0-1 loss function (with respect to $\mathbf{X}'$). Note that for Equation (38) to hold, the number of samples $M$ depends on $\epsilon$, $\delta$, and possibly other parameters in the problem. $M$ can be interpreted as the number of queries required to learn the label assignments up to a small error with high probability.

Since the number of bins $b$ is known to the decoding algorithm, it suffices to consider query functions of the form $h = h' \circ \Pi^{m\times n}$, where $h'\colon [b]^{m\times n} \to \{0,1\}$. Furthermore, we suppose that the data is binned before implementing the learning algorithm. Thus, we can write $\mathcal{A}(\cdot,\cdot) = \mathcal{A}'((\Pi^{m\times n})^M(\cdot), \Pi^{m\times n}(\cdot))$, where $\mathcal{A}'\colon ([b]^{m\times n})^M \times [b]^{m\times n} \to \{0,1\}$. Denote $\overline{\mathbf{X}} = \Pi^{m\times n} \circ \mathbf{X}$ to be the random variable taking values in $[b]^{m\times n}$, obtained by binning the entries of $\mathbf{X}$. Then, Equation (38) can be rewritten as

$$\Pr_{\overline{\mathbf{X}}_1,\ldots,\overline{\mathbf{X}}_M}\left(\Pr_{\overline{\mathbf{X}}'}(\mathcal{A}'((\overline{\mathbf{X}}_i)_{i=1}^M), \overline{\mathbf{X}}') \neq Q'(\overline{\mathbf{X}}')) \leq \epsilon\right) \geq 1 - \delta, \tag{39}$$

For this reason, it suffices to assume that the input data corresponds to the labels of the bins.

### E.1 VC dimension bounds for tabular data using query function of He et al. (2024)

In He et al. (2024), the $z$-score function is of the form $F_z(\mathbf{X}) = \beta_0 + \sum_{j=1}^n \beta_j T_j(\mathbf{X}) + \sum_{j=1}^n \alpha_j T_j(\mathbf{X})^2$, where for all $i,j$, we have $\beta_i, \alpha_j \in \mathbb{R}$ and $T_j(\mathbf{X})$ is the Hamming weight of the label string corresponding to the $j$th column of $\mathbf{X}$. As noted earlier in this section, it suffices to focus our attention to the case where the input is in $[b]^{m\times n}$, obtained by binning the original data. Our goal is to embed all possible query functions into a neural network and use known bounds on the VC dimension for learning the true query function.

**Theorem E.1** (Query complexity bound for querying He et al. (2024) with tabular data)**.** *Given the number of bins $b$, number of rows $m$ and columns $n$ of the query table, there is a neural network that can use*

$$O\left(\frac{mnb\log(mnb)\log(1/\epsilon) + \log(1/\delta)}{\epsilon}\right) \tag{40}$$

*training data (labeled data consisting of query table and whether or not the table is watermarked according to He et al. (2024)), to learn the function which determines whether an input table is watermarked, with error at most $\epsilon > 0$ with respect to 0-1 loss (see Equation (39)), and with probability greater than or equal to $1 - \delta$ over the training samples.*

*Proof.* We begin by embedding the labels $k \in [b]$ into vectors, so as to vectorize the inputs $[b]^{m\times n}$. To that end, associate the label $k \in [b]$ to the standard unit vector $\mathbf{e}_k \in \mathbb{R}^b$, where $\mathbf{e}_k$ is the vector with 1 at the $k$th entry and zero elsewhere. Let $\mathbf{X} \in [b]^{m\times n}$ be the input data, and let it be mapped to the vector $\oplus_{j=1}^n \oplus_{i=1}^m \mathbf{e}_{\mathbf{X}_{ij}}$. Here, the columns of $\mathbf{X}$ are vertically stacked on top of each other after vectorizing.

Let $\mathbf{w} \in \{0,1\}^b$ denote a candidate vector of labels. Observe that $\langle \mathbf{w}, \mathbf{e}_k \rangle = \mathbf{w}_k$ for all $k \in [b]$. Furthermore, for $j \in [n]$, we have $\langle \mathbf{w}^{\oplus m}, \oplus_{i=1}^m \mathbf{e}_{\mathbf{X}_{ij}} \rangle = \sum_{i=1}^m \mathbf{w}_{\mathbf{X}_{ij}}$. If $\mathbf{w}$ is the correct labeling vector for the $j$th column, then $\sum_{i=1}^m \mathbf{w}_{\mathbf{X}_{ij}} = T_j(\mathbf{X})$. By definition, we have $T_j(\mathbf{X}) \in [0,m]$ for all $j \in [n]$ and all $\mathbf{X} \in [b]^{m\times n}$.

Next, we show how to obtain $T_j(\mathbf{X})^2$ from $T_j(\mathbf{X})$. First, add another layer where we map $T_j(\mathbf{X})$ to $(m+1)T_j(\mathbf{X})$, by choosing the weight to be $m+1$. Next, choose the following piecewise polynomial activation function:

$$\psi(z) = \begin{cases} 0 & \text{if } z \leq 0 \\ z & \text{if } z \in (0, m] \\ z^2 & \text{if } z > m. \end{cases} \tag{41}$$

Then, since $T_j(\mathbf{X}) \in [0,m]$, we obtain $\psi(T_j(\mathbf{X})) = T_j(\mathbf{X})$. On the other hand, since $(m+1)T_j(\mathbf{X}) \in \{0\} \cup [m+1, m(m+1)]$, we obtain $\psi((m+1)T_j(\mathbf{X})) = (m+1)^2 T_j(\mathbf{X})^2$. The constant factor of $(m+1)^2$ can be absorbed into the weights in the next layer.

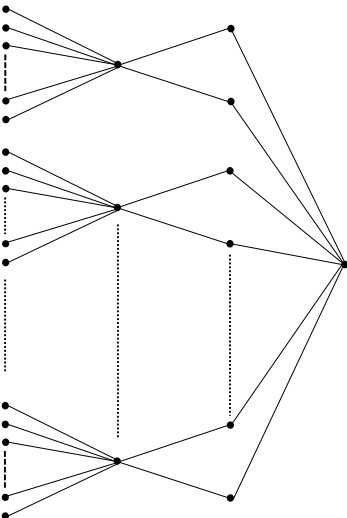

Figure 4: Neural network architecture to embed the problem of learning the query function. The activation function is applied at every node, except the nodes in the input and the output layers.

This motivates us to propose the following neural network architecture for the problem of learning the query function (see Figure 4). The input nodes contain vectorized indices, such that columns of $\mathbf{X}$ are stacked on top of each other. Therefore, there are a total of $mnb$ input nodes. The first hidden layer computes $T_1(\mathbf{X}), \ldots, T_n(\mathbf{X})$. Since the $T_j(\mathbf{X}) \in [0, m]$ for all $j \in [n]$, the activation function acting on the first hidden layer does nothing. The second layer effectively fans out $T_j(\mathbf{X})$ for $j \in [n]$. The top node maps $T_j(\mathbf{X})$ to itself, whereas the bottom node maps $T_j(\mathbf{X})$ to $(m + 1)T_j(\mathbf{X})$. Thus, applying the activation function at the second hidden layer gives us $T_j(\mathbf{X})$ and $(m + 1)^2 T_j(\mathbf{X})^2$ for $j \in [n]$. The output layer is obtained by applying weights and biases to the second hidden layer. Thus, the output of the neural network is of the form $F_z(\mathbf{X}) = \beta_0 + \sum_{j=1}^n \beta_j T_j(\mathbf{X}) + \sum_{j=1}^n \alpha_j T_j(\mathbf{X})^2$. Since we need to compute the VC dimension, we apply the sign (or indicator) function at the output layer. Therefore, the hypothesis class defined by this neural network is the set of functions computed by it over all the weights and biases at each layer, with a sign function (see (37)) applied at the end. By construction, all the query functions are a part of this hypothesis class. Thus, an upper bound on the VC dimension of the hypothesis class defined by the neural network also gives an upper bound on the VC dimension of the hypothesis class defined by all the query functions.

There are a total of $O(mnb)$ weight and bias parameters before the first layer, $O(n)$ parameters before the second layer, and finally, $O(n)$ parameters before the output layer. Thus, we have a total of $\mathrm{wt} = O(mnb)$ parameters. We have a total of $L = 3$ layers. Let $\mathrm{wt}_i$ denote the number of weight and bias parameters from the input layer till the $i$th layer. Then, the effective depth of the neural network is $\overline{L} = \sum_{i=1}^L \mathrm{wt}_i/\mathrm{wt}$ is of order 1. Then, by (Bartlett et al., 2019, Theorem 6), the VC dimension of the neural network hypothesis class is bounded above by $O(mnb \log(mnb))$. Therefore, by (Shalev-Shwartz & Ben-David, 2014, Theorem 6.8), we can infer that

$$M = O\left(\frac{mnb \log(mnb) \log(1/\epsilon) + \log(1/\delta)}{\epsilon}\right) \tag{42}$$

training samples are sufficient to learn the query function to within an error of $\epsilon$ (with respect to the $0 - 1$ loss) with probability greater than or equal to $1 - \delta$ over the training samples.

Now, denote $\mathcal{A}'(\text{training data}, \cdot)$ to be the query function that is learned from the training data. Then, if the distribution of the data is uniform, then $\Pr_{\overline{\mathbf{X}}'}(\mathcal{A}'(\text{training data}, \overline{\mathbf{X}}') \neq Q'(\overline{\mathbf{X}}'))$ is equal to the fraction of indices where the learned query function differs from the true query function. As a result, setting $\epsilon = 0.5/(mnb)$, we obtain the situation where we learn the query function exactly. Thus, we need $O((mnb)^2 \log(mnb) \log(b) +$

$mnb \log(1/\delta))$ samples to learn the query function exactly with probability at least $1 - \delta$ over the training samples. □

## E.2 VC dimension bounds for tabular data using query function of Algorithm 1

Unlike He et al. (2024), in our work, we only watermark $n$ columns of the input table (see Algorithm 1). Thus, during query time, one would need to identify first the columns of the query table that are watermarked (referred to as value columns). As stated in Section 5, the query function returns 1 if the $z$-score of each of the value columns are greater than an input threshold $z_{\text{th}} \geq 0$. Thus, we prove the complexity of decoding Algorithm 1 for $\alpha \leq 0.5$ (which corresponds to $z_{\text{th}} \geq 0$).

**Theorem E.2** (Query complexity bound for decoding Algorithm 1). *Given the number of bins $b$, number of rows $m$ and columns $2n$ of the query table, and assuming that the significance level $\alpha$ for $z$-test in Section 5 is at most $0.5$, there is a neural network that can use*

$$O\left(\frac{mnb \log(mnb) \log(1/\epsilon) + \log\left(1/\delta\right)}{\epsilon}\right) \tag{43}$$

*training data (labeled data consisting of query table and whether or not the table is watermarked according to Algorithm 1), to learn the function which determines whether an input table is watermarked, with error at most $\epsilon > 0$ with respect to 0-1 loss (see Equation (39)), and with probability greater than or equal to $1 - \delta$ over the training samples.*

*Proof.* Following the proof of Theorem E.1, we embed data $\mathbf{X} \in [b]^{m \times 2n}$ before feeding into a neural network. First, we embed the labels $k \in [b]$ into vectors, so as to vectorize the inputs $[b]^{m \times 2n}$. To that end, associate the label $k \in [b]$ to the standard unit vector $\mathbf{e}_k \in \mathbb{R}^b$, where $\mathbf{e}_k$ is the vector with 1 at the $k$th entry and zero elsewhere. Let $\mathbf{X} \in [b]^{m \times 2n}$ be the input data, and let it be mapped to the vector $\oplus_{j=1}^{2n} \oplus_{i=1}^{m} \mathbf{e}_{\mathbf{X}_{ij}}$. Here, each column of each row of $\mathbf{X}$ are vertically stacked on top of each other after vectorizing.

Now, our goal is to show that there is a neural network architecture with appropriate weights, biases, and activation function such that the output of the neural network is the true query function. The true query function outputs 1 *if and only if* the $z$-score of all the *value* columns is above the threshold $z_{\text{th}}$. As per Section 5, we denote the $z$-score of the $j$-th column as $z_j = 2(T_j(\mathbf{X})/\sqrt{m}) - \sqrt{m}$, for all $j \in [2n]$. For obtaining the true query function, we need to know which columns correspond to the value columns as well as which bins are watermarked in a given value column. Since we seek to embed the true query function into a neural network, we assume that we know the indices $\mathcal{V}$ corresponding to the value column as well as the true labeling vectors. Algorithm 1, by design, ensures that $\mathcal{V} \subseteq [2n]$ and $|\mathcal{V}| = n$.

The first hidden layer contains a total of $2n$ nodes. Let $\mathbf{w} \in \{0,1\}^b$ denote a candidate vector of labels (1 is interpreted as watermarked, while 0 is interpreted as not watermarked). Observe that $\langle \mathbf{w}, \mathbf{e}_k \rangle = \mathbf{w}_k$ for all $k \in [b]$. Furthermore, for all $j \in [n]$, we have $\left\langle \mathbf{w}^{\oplus m}, \oplus_{i=1}^{m} \mathbf{e}_{\mathbf{X}_{ij}} \right\rangle = \sum_{i=1}^{m} \mathbf{w}_{\mathbf{X}_{ij}}$. If $j \in \mathcal{V}$ and $\mathbf{w}$ is the correct labeling vector for the $j$th column, then $\sum_{i=1}^{m} \mathbf{w}_{\mathbf{X}_{ij}} = T_j(\mathbf{X})$. Since $T_j(\mathbf{X}) \in [0, m]$, we add a bias of $\sqrt{m} + 1$ so that $T_j(\mathbf{X}) + \sqrt{m} + 1 > \sqrt{m}$. If, on the other hand, if $j \notin \mathcal{V}$, i.e., we have a key column, then we set the weight $\mathbf{w}$ equal to the zero vector and add a bias of $m + \sqrt{m} + 2 > \sqrt{m}$. (We choose such a bias to internally distinguish a key column from a value column, since $T_j(\mathbf{X}) + \sqrt{m} + 1 < m + \sqrt{m} + 2$.) Subsequently, we apply the following piecewise-linear activation function:

$$\psi(x) = \begin{cases} 0 & \text{if } x \leq 0 \\ 1 + \frac{1}{2n} & \text{if } 0 \leq x \leq \sqrt{m} \\ x & \text{if } x > \sqrt{m}. \end{cases} \tag{44}$$

Then, the output of the first hidden layer (after applying the weights, biases, and activation function) is equal to $T_j(\mathbf{X}) + \sqrt{m} + 1$ if $j \in \mathcal{V}$, while it is equal to $m + \sqrt{m} + 2$ if $j \notin \mathcal{V}$.

For the second layer, our goal is to obtain the $z$-scores for the value columns and compare it to the threshold. To achieve this, we add a weight of $2/\sqrt{m}$ and a bias of $-\sqrt{m} - 2 - 2/\sqrt{m} - z_{\text{th}}$. Applying such a weight an bias to the value column $j \in \mathcal{V}$ gives $(2/\sqrt{m})(T_j(\mathbf{X}) + \sqrt{m} + 1) - \sqrt{m} - 2 - 2/\sqrt{m} - z_{\text{th}} = z_j - z_{\text{th}}$, where $z_j$

is the $z$-score of the $j$th column. Now, if $z_j - z_{\text{th}} \leq 0$ (or $z_j \leq z_{\text{th}}$), then the activation function outputs 0. On the other hand, if $z_j - z_{\text{th}} > 0$, then since $z_j \leq \sqrt{m}$ and $z_{\text{th}} \geq 0$ (as $\alpha \leq 0.5$ by assumption), we have $z_j - z_{\text{th}} \leq \sqrt{m}$, so that the activation function outputs $1 + 1/(2n)$. Note that for the *key* columns, we can set the weights as 1 and biases as 0. Then applying the activation function to these nodes, we have the output $m + \sqrt{m} + 2$ as before.

Finally, in the third layer (which is also the final/output layer), for each column corresponding to the values, we apply a weight of 1 and a bias of $-n$. For the key columns, we apply weight and bias equal to 0 (which is equivalent to ignoring the key columns at the final layer). Therefore, the output at the final layer is $\sum_{j \in \mathcal{V}} (1 + 1/(2n)) \mathbf{1}[z_j - z_{\text{th}} > 0] - n$, where $\mathbf{1}[A]$ denotes the indicator function of the event $A$ (i.e., $\mathbf{1}[A] = 1$ if $A$ is true and 0 otherwise). It can be verified that the output is positive if and only if $z_j > z_{\text{th}}$ for all $j \in \mathcal{V}$. Then, we apply the sign function given in (37) to this output. Thus, such a network can learn the true query function required. A schematic of the proposed neural network architecture is given in Figure 5.

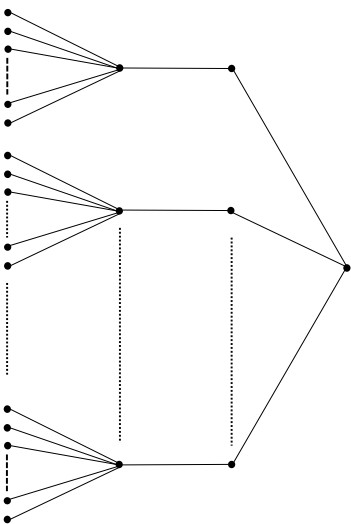

Figure 5: Neural network architecture to embed the problem of learning the query function for dataset watermarked according to Algorithm 1. The activation function is applied at every node, except the nodes in the input and the output layers.

The total number of parameters required to represent the weights and biases before the first layer is $O(mnb)$, before the second layer is $O(n)$, and before the third layer is $O(n)$. Thus we have a total of $O(mnb)$ parameters. We also have 3 layers in the architecture. The effective depth of our neural network is $O(1)$. Then, using (Bartlett et al., 2019, Theorem 6), the VC dimension of the neural network hypothesis class is bounded above by $O(mnb \log(mnb))$. Using (Shalev-Shwartz & Ben-David, 2014, Theorem 6.8),

$$M = O\left( \frac{mnb \log(mnb) \log(1/\epsilon) + \log(1/\delta)}{\epsilon} \right) \tag{45}$$

training samples are sufficient to learn the query function up to error $\epsilon$ with respect to the $0-1$-loss with probability greater than or equal to $1 - \delta$ over the training samples.

Now, denote $\mathcal{A}'(\text{training data}, \cdot)$ to be the query function that is learned from the training data. Then, if the distribution of the data is uniform, then $\Pr_{\overline{\mathbf{X}}'}(\mathcal{A}'(\text{training data}, \overline{\mathbf{X}}') \neq Q'(\overline{\mathbf{X}}'))$ is equal to the fraction of indices where the learned query function differs from the true query function. As a result, setting $\epsilon = 0.5/(mnb)$, we obtain the situation where we learn the query function exactly. Thus, we need $O((mnb)^2 \log(mnb) \log(b) + mnb \log(1/\delta))$ samples to learn the query function exactly with probability at least $1 - \delta$ over the training samples. $\qquad\square$

**Corollary E.3** (Query complexity of row queries to both He et al. (2024) and Algorithm 1)**.** *For queries of the form* $\mathbf{X} \in \mathbb{R}^{m \times 1}$ *and query function that computes* $z = 2T(\mathbf{X})/\sqrt{m} - \sqrt{m}$ *or any linear function of* $T(\mathbf{X})$*, there is a neural network that can learn this query function up to error* $\epsilon$ *in* 0-1 *loss using* $O\left(\frac{mb\log(1/\epsilon) + \log(1/\delta)}{\epsilon}\right)$ *samples with probability* $1 - \delta$ *over the training samples.*

*Proof.* Using $n = 1$ in Theorem E.2 gives us the result. For He et al. (2024), observe that when a single row is input, $T_j(\mathbf{X})$ for all $j \in n$ is in $\{0, 1\}$. As such, the statistical test uses $z$-score tests on the standard normal distribution, and not the $\chi^2$ test. The hypothesis class corresponding to the neural architecture as defined in Theorem E.2 is thus PAC learnable for this problem, and so the query complexity bound follows by setting $n = 1$. $\qquad\square$

### E.3 A lower bound for row queries to He et al. (2024) given the $z$-scores

Consider the case that the continuous space in $(0, 1]$ is uniformly discretized using bins of size $1/b$, i.e., construct bins in $(0, 1]$ as $-\left\{(0, \frac{1}{b}], (\frac{1}{b}, \frac{2}{b}], \ldots, (\frac{b-1}{b}, 1]\right\}$. Then each of these bins are labeled $\{0, 1\}$ with some probability $1 - p$ and $p$ respectively. We save this data-structure, and allow anyone to query it with $S \in (0, 1]^m$, responding with the $z$-score (Equation 3) and whether the data is watermarked or not. In this section, we want to bound the minimum number of queries one can make to the model and estimate the watermarking scheme with high accuracy. Specifically we ask:

*When the number of bins $b$ is known, what is the query complexity to correctly identify all red and green intervals used for watermarking?*

Observe that for a row, the $z$-score is computed as: $2\sqrt{m}\left(\frac{T}{m} - \frac{1}{2}\right)$, where $T$ is the number of values in the row falling in the bins with label 1.

When $b$ is known to the adversary, the problem can be reduced to the 2-color Mastermind game. In 2-color mastermind the codemaker creates a secret code using a sequence of colored pegs, and the codebreaker guesses the sequence. At each query by the codebreaker, the codemaker provides the number of pegs they correctly guessed. As such, given $z$ is the output of each query, one can easily compute the number of values in each bin $\{0, 1\}$. Thus, given any $b$, one can reduce this problem to the 2-color Mastermind game. This specific form of Mastermind has been extensively studied in the literature Chvátal (1983); Knuth (1976). In particular, with 2 possible colors: red and green, the maximum number of row queries needed to recover the exact watermarking scheme is $\Theta\left(\frac{b\log(2)}{\log(b)}\right)$. Hence, the 'red-green' watermarking scheme by He et al. (2024) can be learned by an adversary making $\Theta\left(\frac{b\log(2)}{\log(b)}\right)$ queries to the detector. Note that this lower bound does not directly apply to the upper bounds derived using VC theory in Appendix E.1 and Appendix E.2, since here we assume access to the $z$-score for each query.

