# OpenReview forum: "Adaptive and Robust Watermark for Generative Tabular Data"
_TMLR — Rejected by TMLR_

### Review · Reviewer_e8JR · 2025-06-20

**Summary Of Contributions:**

This paper studies a watermarking scheme for tabular data that partition the feature space into pairs of(key, value) columns and embeded secret signal into the value columns using a method similar to existing works. In particular, the selection of the green/red list positions for embedding is determined by the information carried in the key columns.

**Audience:**

Yes

**Claims And Evidence:**

Yes

**Requested Changes:**

The manuscript would be considerably strengthened by addressing the weaknesses outlined above. Furthermore, it would be great to expand the discussion of the underlying motivation for watermarking. For example, consider its potential role in preventing phenomenon such as model collapse [1].



**References**
[1] Shumailov, I., Shumaylov, Z., Zhao, Y., Papernot, N., Anderson, R. and Gal, Y., 2024. AI models collapse when trained on recursively generated data. Nature, 631(8022), pp.755-759.

**Strengths And Weaknesses:**

**Strength**

(1) The method is straightforward to follow.

(2) The effectiveness of the method is supported by both theoretical results and experiments.

**Weakness**: There are some serious theoretical flaws in this paper, which undermines the overall integrity of the paper.

**(1) Issues with Theorem 4.1.** Theorem 4.1 is difficult to understand because it mixes an \emph{expectation} guarantee with a \emph{high-probability} guarantee without clearly separating the two. In addition, the proof’s upper bound for $E[|x-x_w|]$
includes the term $\frac{2b}{b 2^k}=2\delta$ which does not vanish as $b \to \infty$. Hence the stated result in Theorem 4.1 appears to be incorrect.

**(2) Issues with Lemma 5.1.** The proof of Lemma 5.1 seems not to be correct. In the assumption of Lemma 5.1, $b \to \infty$ is needed. However, this condition is never used in the proof for it. Also, is Lemma 5.1 really true for any probability distribution without continuous assumptions?

**(3) Overstated Robustness Claims**: The paper claims that it is more robust than some previous works such as [1]. However, this seems to be overclaimed.

**(4) Robustness to Feature Selection:**  The paper asserts robustness against feature-selection attacks, yet this claim offers no clear advantage over prior work such as [1]. In fact, [1] embeds watermarks without relying on a key column to determine the green/red list construction and is therefore already less susceptible to feature-selection attacks.

**(5) Robustness to Truncation**: The paper claims robustness to truncation attacks, yet it is unclear how this robustness surpasses that of the prior schemes proposed in [1] and [3]. In fact, the guarantees provided by these earlier methods appear to be at least as strong as those offered here.

**(6) Robustness to Additive Noise Attacks:** The paper offers theoretical guarantees only for additive noise that is uniformly bounded, yet it omits more realistic perturbations, such as Gaussian noise studied in recent work [1,2]. A broader, systematic analysis of robustness under diverse additive-noise attacks is still required.

**(7) Clarity of Novelty and Proper Attribution:** Several key results in the manuscript appear to closely parallel earlier findings reported in [1,3]. For example, Theorem 4.1 is similar to Theorem 1 in [1]. Corollary 4.2 is similar to  Corollary 1.1 in [1]. Lemma 5.1 is similar to Lemma 1 in [1]  The paper would benefit from more rigorous citation practices—explicitly indicating which theorems, proof techniques, and robustness bounds are adapted from prior work and which constitute new contributions. Clear attribution will help readers discern the novel insights offered by this study.

**References**

[1] He, H., Yu, P., Ren, J., Wu, Y.N. and Cheng, G., 2024. Watermarking generative tabular data. arXiv preprint arXiv:2405.14018.

[2] Zhu, C., Tang, J., Galjaard, J.M., Chen, P.Y., Birke, R., Bos, C. and Chen, L.Y., 2025. Tabwak: A watermark for tabular diffusion models. In International Conference on Learning Representations (pp. 1-28). OpenReview. net.

[3] Zheng, Y., Xia, H., Pang, J., Liu, J., Ren, K., Chu, L., Cao, Y. and Xiong, L., 2024, December. Tabularmark: Watermarking tabular datasets for machine learning. In Proceedings of the 2024 on ACM SIGSAC Conference on Computer and Communications Security (pp. 3570-3584).

---

> ### Author Response · Authors · 2025-08-15
>
> We thank the reviewer for the thoughtful comments. Please find our responses to specific questions below:
>
> - Issue with Theorem 4.1: The current theorem statement in Theorem 4.1 is confusing, as correctly pointed out by the reviewer. We have revised the theorem statement in the revision manuscript (highlighted in blue) and the corresponding proof (Appendix A). Now, the theorem statement should only read: with high probability, the $L_\infty$ distance between the watermarked and original dataset is upper bounded by a term that scales with $\frac{1}{b}$.
>
> - Issue with Lemma 5.1: We have revised the theorem statement and the proof of Lemma 5.1 to remove the assumption that $b \rightarrow \infty$. Lemma 5.1 is indeed true for any probability distribution, we do not need to assume that the distribution is continuous.
>
> - Overstated robustness claims: We have revised the language of the comparison to prior work of He et al., [2024] and Zheng et al., [2024] in the related work section (Section 2). In summary, compared to He et al., [2024], our approach has the same theoretical robustness as theirs against feature-selection and truncation, but is more robust to the simple spoofing attack (described in Section 8.2). In addition, we have added theoretical analysis of robustness to additive Gaussian noise in Appendix C.1, and a meta-theorem on robustness against additive noise drawn from \textbf{any} distribution satisfying a mild condition.
>
>     \item Clarity of novelty: We want to emphasize that the similarities in Theorem 4.1, Corollary 4.2 and Lemma 5.1 in our manuscript to prior work [1,3] are only due to the shared problem setting. In this work, we aim to watermark generative tabular data while maintaining high data fidelity (Theorem 4.1, Corollary 4.2), and the watermark detection follows statistical hypothesis testing (Lemma 5.1). However, our watermark embedding technique is different from that of prior work, leading to different theoretical guarantees and proof techniques. Particularly, the proof techniques in He et al., [2024] and Zheng et al., [2024] cannot be applied to our work since (1) we do not force red and green intervals to always be next to each other (as in [1]), and hence the fidelity guarantee only hold with high probability for our algorithm and (2) the data in our table is assumed to be in $[0,1]$ instead of a custom range $[-p, p]$, leading to different robustness to additive noise analysis. Furthermore, due to our columns pairing scheme, we use one-proportion $z$-test with Bonferroni correction to detect our watermark, which is different from prior work. A summary of these differences can be found in Table 1 (Section 2).
>
> - Motivation for watermarking: We have added model collapse as a potential motivation for using watermarking in the introduction.

---

### Review · Reviewer_kYCd · 2025-07-08

**Summary Of Contributions:**

In short, the paper proposes a watermarking technique specifically designed for continuous tabular data. The major contribution of this manuscript is to propose a hashing mechanism to augment the existing continuous tabular data watermarking procedure. The proposed method is presented with proofs for major claims and experiment results on both synthetic tabular data and generative tabular data (namely, tabular data sampled from pretrained generative models).

**Audience:**

Yes

**Broader Impact Concerns:**

NaN

**Claims And Evidence:**

Yes

**Requested Changes:**

Please refer to the weaknesses section to make necessary changes or clarification of the original manuscript to secure my recommendation for acceptance.

**Strengths And Weaknesses:**

**Strengths**:
+ The idea of using a hashing mechanism to augment the existing procedure is potentially useful and could be theoretically interesting.
+ The experiments covered both synthetic data and generative tabular data, which could be potentially helpful for practical concerns.

**Weaknesses**
+ **(Possible) major technical flaws in the proofs of major theoretical results**
    + The major theoretical result on fidelity, thm 4.1, seems problematic or at least rather hard to understand, if I may say so. Specifically in the appendix A, proof of thm 4.1 section, I kindly request the authors to further clarify the following parts if possible:
         + Can the authors comment on or further clarify the definition of the expectation used in defining the element-wise distance?
         + Can the authors further elaborate on the definition and introduction of the variable `k` used in the proof? Is `k` a fixed value? If so, the conclusion is very weird as it is saying that: with a **fixed** probability $1 - \delta$, the final inequality holds; a **fixed** portion, i.e., $\delta$, of times, the inequality will be violated.
         + If `k` is not a fixed value and goes to infinity (which means infinitely large searching radius I assume?), maybe most importantly, can the authors clarify what exactly is the concentration inequality used for deriving the final inequality result? Specifically, what is the assumption of the data distribution we are inspecting? Maybe I missed something, but I failed to see how the union bound technique helps to translate to the final result. It would be very helpful if the authors could help with this issue.
    + Following thm 4.1, the authors further derive the corollary 4.2. In the proof presented in the appendix, it is not very clear to me how to translate the inequality **with expectation** from thm 4.1 to the inequality from the 3rd line to the 4th line **without expectation** in the proof.
    + The core part or the foundation of the detectability is based on Lemma 5.1. However, I found the proof of Lemma 5.1 also very confusing to me. I would appreciate it if the authors could clarify the following parts:
        + How to obtain the 4th line in the proof from the 3rd line? It remains unclear to me how to transfer the conditional probability in the 3rd line to the marginal probability in the 4th line.
        + Does the proof of Lemma 5.1 imply that we can always get the 1/2 probability even with $b=1$? If so, why do we need to assume that $b \to \infty$?
        + What is the difference between Lemma 5.1 and Lemma 1 from [He et al., 2024], which is the major competitor of the proposed method?
    + I did not carefully check the remaining proofs due to the time limit. But I think the possible major technical flaws (or at least ambiguities) of the theoretical foundation regarding the fidelity and detectability of the proposed method have made it necessary for a major revision at least.
+ **Lack of appropriate treatment and citation of the theoretical results**.
    + Following the previous question, I would appreciate it if the authors could clarify the major difference between Lemma 5.1 and Lemma 1 from [He et al., 2024]. It seems to me that these two theoretical results are rather similar, if not identical, while the authors re-state the Lemma 5.1 from Lemma 1 from [He et al., 2024] without proper citation or discussion of the previous work.
+ **Insufficient experiment evidence for the effectiveness**
    + If I understand correctly, the core motivation of augmenting the existing watermarking procedure proposed in [He et al., 2024] is to further strengthen the robustness against attacks. However, while the authors claim and propose certain robustness guarantees for the proposed method, there is no direct comparison, either theoretically or empirically, between the proposed method and existing methods, except for the synthetic spoofing experiment mentioned in sec 8.2.
    + However, for the spoofing experiment, it
        + a) requires both a clean sample and a watermarked sample, while in practice it seems to be a chick-n-egg problem: how to identify the watermarked sample as an adversary in the first place, without knowing the exact underlying watermarking mechanism?
        + b) it successfully lower the p_value to under .05 level when replacing ~40% of the original data. This leads to at least two following questions:
            + i) how to make sure, quantitatively, that the original data is not severely distorted?
            + ii) What if the detector simply chooses a much lower p_value threshold (significance level), say, .005 or even .001 to minimize the false positive rate?
    + An additional, maybe minor, issue is that for the results presented in fig. 2(d), the fidelity is measured in mse, while the theoretical results are based on the $\mathcal{l}_\infty$ norm and wasserstein distance.

Hengzhi He, Peiyu Yu, Junpeng Ren, Ying Nian Wu, and Guang Cheng. Watermarking generative tabular data, 2024. URL https://arxiv.org/abs/2405.14018.

---

> ### Author Response · Authors · 2025-08-15
>
> We thank the reviewer for their thoughtful comments. Please find our responses to specific questions below:
>
> - Theorem 4.1 and Corollary 4.2: We have revised the theorem statements and corresponding proofs (Appendix A) on fidelity to add more clarity. Please see the changes in the revised manuscript (in blue).
>
> - Lemma 5.1: We have revised the proof of Lemma 5.1 (Appendix B). Please see the changes in the revised manuscript (in blue). We can always get the $\frac{1}{2}$ probability for any $b$ and does not need to assume that $b \rightarrow \infty$. While Lemma 5.1 and Lemma 1 from He et al., [2024] has similar statement, the way we embed watermark into tabular data is different from their algorithm. Particularly, we do not need to force red and green intervals to be next to each other. Hence, the proof technique for Lemma 5.1 is also different from that of He et al., [2024]'s Lemma 1.
>
> - Citation of theoretical results: We want to emphasize that due to differences in watermark generation algorithms, the proofs of our Lemma 5.1 and He et al., [2024]'s Lemma 1 are different and incompatible. The lemma statements are similar because we also want to split the interval $[0,1]$ into half green intervals and half red intervals, without enforcing red and green intervals to be next to each other.
>
> - Insufficient experiment evidence: we follow standard spoofing attack in the literature of watermark stealing [Jovanovic et al., 2024], and assume that the adversary knows which watermark mechanism is used on the tabular dataset. We spoof the watermark by only changing the decimal points of the original data to the nearest values in the watermarked dataset. Hence, large enough watermarked dataset and sufficiently diverse watermark samples, our spoofing approach does not distort the original data by much. If the detector simply chooses a much lower $p$ value threshold, our watermark is still robust against this spoofing attack (Figure 3a), while He et al., [2024]'s watermark can still be spoofed as the $p$ value approaches close to $0$ after 60% of the data is processed (Figure 3b). When the entire dataset is replaced in this manner, we can successfully spoof He et al., [2024]'s watermark with probability $1$, since the detector in their algorithm will observe all samples in our spoofed dataset as being in green intervals. We report MSE as a measure of fidelity in downstream task in our experiment instead of norm and Wasserstein distance to follow the tradition of prior work [He et al., 2024, Zheng et al., 2024}.

---

### Review · Reviewer_RdsV · 2025-08-01

**Summary Of Contributions:**

This paper introduces a method for the watermarking of generative tabular data. Specifically, this work presents a fine-grained watermarking scheme for tabular data with strong statistical guarantees and desirable properties. Moreover, this author also introduces some approaches to ensure the robustness and high-fidelity data quality for the downstream tasks.

**Audience:**

Yes

**Claims And Evidence:**

No

**Requested Changes:**

NA

**Strengths And Weaknesses:**

**Strengths:**
1. This work discusses some new ways to watermark the generative tabular dataset.
2. Some preliminary experiments are also conducted to validate the effectiveness of the proposed approach.
3. I

**Weakness:**
1. The motivations of this work are not clear. The authors do not clearly explain why they choose to watermark the generative tabular data.
2. The technical contribution of this paper seems to be incremental. There is no obvious technical contributions to secure this paper accepted by a top journal.
3. The experiments of this work also have some problems. The comparison with the SOTA is still not enough.

---

> ### Author Response · Authors · 2025-08-15
>
> We thank the reviewer for their thoughtful comments. Please find our responses to specific questions below:
>
> - Motivation of this work is not clear: We have added 'model collapse' as a real-life motivation of watermarking in the introduction of the revised manuscript (Section 1).
>
> - Technical contributions: We want to emphasize that our theoretical contributions are not incremental, as our watermarking technique presents new challenges in the analysis not yet addressed by prior work. We have since revised the proofs of our theoretical claims to make this distinction clear. Please see the revised manuscript and the changes (highlighted in blue).
>
> - Experiments: We have added the comparisons to state-of-the-art post-process watermarking techniques in the appendix of the revised manuscript. In summary, our method can both obtain minimal distortion in downstream task (measured by MLE Gap) and can be reliably detected under most threat models, whereas other watermarking methods are more susceptible to either table level attacks or cell level attacks. Can the reviewer kindly clarify what problems in our experiments that they are referring to in this comment?

---

### Review · Reviewer_z7Kc · 2025-08-01

**Summary Of Contributions:**

This paper proposed a watermarking approach for tabular data.

**Audience:**

Yes

**Claims And Evidence:**

No

**Requested Changes:**

Please address the weaknesses

**Strengths And Weaknesses:**

This paper proposed a watermarking approach for tabular data. It is recommended to be rejected due to the following reasons:

The lack of novelty. The proposed approach is, in principle, quite similar to the related work "Yihao Zheng, Haocheng Xia, Junyuan Pang, Jinfei Liu, Kui Ren, Lingyang Chu, Yang Cao, and Li Xiong. Tabularmark: Watermarking tabular datasets for machine learning, 2024.". However, this related work is not comprehensively compared in the experiments or discussed in detail to discriminate from the proposed approach. The similarity lies in the fact that: a) both methods uses randomly partitioned green red domains to embed watermark. b) both methods uses one-proportion z-test to detect watermark. Even the proof of robustness share similar key idea.

Lack of comparison to state-of-the-art. There are many related works in database watermarking and tabular dataset watermarking. To name a few, a) Zhu, Chaoyi, Jiayi Tang, Jeroen M. Galjaard, Pin-Yu Chen, Robert Birke, Cornelis Bos, and Lydia Y. Chen. "Tabwak: A watermark for tabular diffusion models." In International Conference on Learning Representations, pp. 1-28. OpenReview. net, 2025.

b) Fang, Liancheng, Aiwei Liu, Henry Peng Zou, Hengrui Zhang, and Philip S. Yu. "RINTAW: A Robust Invisible Watermark for Tabular Generative Models." In The 1st Workshop on GenAI Watermarking.

c) Che, Xin, Mohammad Akbari, Shaoxin Li, David Yue, Yong Zhang, and Lingyang Chu. "Primary Key Free Watermarking for Numerical Tabular Datasets in Machine Learning." In International Conference on Pattern Recognition, pp. 254-270. Cham: Springer Nature Switzerland, 2024.

In summary, this paper is recommended to be rejected.

---

> ### Author Response · Authors · 2025-08-15
>
> We thank the reviewer for their thoughtful comments. Please find our responses to specific questions below:
>
> - Lack of novelty: We respectfully disagree with the reviewer's assessment that our watermarking approach is similar to that of TabularMark. Our watermark specifically divides the dataset into pairs of (key, value) columns, and embed the watermark into all value columns using the corresponding key columns as seeds to a random function. TabularMark only embeds the watermark into a few key cells of a specific column (often chosen to be the label column). At a high level, our watermarking technique leverages the structure of the tabular dataset, which is not present in TabularMark's method. The idea of randomly partitioning green and red intervals are also different in our approach: we define these intervals on the $[0,1]$ range and modify cells in the value columns to be in green intervals, while TabularMark define the partitions on a perturbation range $[-p, p]$ and inject these perturbations into the selected key cells. Finally, our watermark use one-proportion $z$-test with Bonferroni correction to detect the watermark due to our pairing scheme, while TabularMark only uses one-proportion $z$-test with no modification.
>
>     The proof of robustness for our method initially share the same idea of calculating the expected number of modified cells that still allow the watermark to be detected afterward. However, since our watermarking technique only embed the watermark into fractional part (a range of $[0,1]$), the analysis is different compared to TabularMark. We have clarified this difference in the new proof, updated in our revised manuscript.
>
> - Lack of comparison to state-of-the-art: We thank the reviewer for directing us to these related work in tabular watermarking. We have since added a discussion of these related work in Section 2, and empirically compare our performance to WGTD [He et al., 2024], TabularMark [Zheng et al., 2024] and RINTAW [Fang et al., 2025] in the appendix D.3. In summary, our method can both obtain minimal distortion in downstream task (measured by MLE Gap) and can be reliably detected under most threat models, whereas other watermarking methods are more susceptible to either table level attacks or cell level attacks.

---

### Decision · Action_Editor_Zip1 · 2025-09-05

**Recommendation:** Reject

**Audience:**

No

**Audience Explanation:**

All reviewers reach the consensus that there is limited technical contribution in this submission compared to past work. The authors did not manage to explain non-trivial differences in the rebuttal. Several reviewers are also concerned about unclear motivation.

**Claims And Evidence:**

No

**Claims Explanation:**

Several reviewers are concerned about the lack of experimental comparison to SOTA baselines. The reviewers' rebuttal did not adequately address this concern.